# SkillWrapper:
# Generative Predicate Invention for
# Skill Abstraction

## Abstract

Generalizing from individual skill executions to solving long-horizon tasks remains a core challenge in building autonomous agents. A promising direction is learning high-level, symbolic representations of the low-level skills of the agents, enabling reasoning and planning independent of the low-level state space. Recent advances in foundation models have made it possible to generate symbolic predicates that operate on raw sensory inputs—a process we call *generative predicate invention*—to facilitate downstream representation learning. However, it remains unclear *which* formal properties the learned representations must satisfy, and *how* they can be learned to guarantee these properties. In this paper, we address both questions by presenting a formal theory of generative predicate invention for skill abstraction, resulting in symbolic operators that can be used for provably sound and complete planning. Within this framework, we propose SkillWrapper, a method that leverages foundation models to actively collect robot data and learn human-interpretable, plannable representations of black-box skills, using only RGB image observations. Our extensive empirical evaluation in simulation and on real robots shows that SkillWrapper learns abstract representations that enable solving unseen, long-horizon tasks in the real world with black-box skills.

## 1 Introduction

An autonomous agent operating in the real world must process low-level sensory and motor signals while reasoning about high-level objectives (Doncieux et al., 2018; Konidaris, 2019). Analogous to how humans can perform complex tasks, like cooking or cleaning, without reasoning about muscle-level control, agents should have internal models of their skills that abstract away nuanced activities on the lower level. Such models must capture the necessary conditions for a skill to be executed (e.g., *"pouring a teapot requires holding it first"*) and the consequences of doing so (e.g., *"pouring a teapot leaves it empty"*). These two properties, known as *preconditions* and *effects* in the AI planning literature, enable compositional reasoning to identify long-horizon plans that can sequence lower-level skills to solve a task. Typically, these models must be specified manually. However, in real-world settings such skill representations may be nontrivial to acquire due to complex inter-skill constraints specific to the agent's embodiment. This calls for algorithms that learn symbolic transition models of black-box skills without hand specification, enabling agents to directly utilize those skills to solve long-horizon tasks with off-the-shelf AI planners.

Traditional approaches of skill abstraction often require factorizing the low-level state space to learn classifiers for each symbolic representation, relying heavily on hand-collected transition data (Konidaris et al., 2018). Recently, foundation models have enabled a new paradigm: generating semantically meaningful predicates directly from raw observations and directly evaluating their truth values on low-level observations (e.g., RGB images)—a process we refer to as *generative predicate invention*. Recent work has explored how foundation models can be used for predicate invention, by generating Python code to implement predicates (Liang et al., 2025) or sampling large predicate pools followed by sub-selection. However, these methods produce ad-hoc planning representations that cannot be guaranteed to solve a given task, and leave core questions on predicate invention unanswered: *what* properties should these learned abstractions satisfy, and *how* can they be learned to achieve these properties?

Figure 1: **Overview of SKILLWRAPPER.** For an agent equipped with black-box skills, SKILLWRAPPER learns skill representations that are compatible with off-the-shelf planners. These representations are comprised of predicates invented by the foundation model. Given a novel planning problem described using the initial state and goal state as RGB images, a foundation model produces the corresponding abstract states by applying the invented predicates to the low-level states. SKILLWRAPPER is *agnostic* to the agent, and we illustrate both real-world (robots) and simulated agents in this figure.

Our answer is twofold. First, we develop a formal theory of generative predicate invention for skill abstraction, precisely characterizing the conditions under which a learned skill representation will be provably *sound* and *complete* with respect to downstream planning. Building on this foundation, we introduce SKILLWRAPPER, a method explicitly designed to guarantee these theoretical criteria. SKILLWRAPPER uses foundation models in three ways: interactively collecting data in the environment, proposing predicates when the current model fails, and classifying predicate truth values based solely on RGB image observations. Using these data and predicates, SKILLWRAPPER learns symbolic representations of black-box skills that are both human-interpretable and directly usable for AI planning.

We highlight the following contributions: (1) A formal theory of generative predicate invention for *provably sound and complete* skill abstraction; (2) SKILLWRAPPER, a principled system built on this framework that leverages foundation models to learn interpretable symbolic representations of black-box skills; and (3) an extensive empirical evaluation of the system, demonstrating effectiveness in simulation and on two real robots.

## 2 PROBLEM SETTING

In this section, we briefly discuss our problem setting while defining it formally in Appendix A. We consider an agent equipped with a finite set of *object-centric skills* $\Omega$, modeled as black-box options (Sutton et al., 1999). The agent can execute any $\omega \in \Omega$ and determine whether it succeeds, but it does not possess a symbolic transition model of these skills. Without such a model, the agent cannot plan over a long horizon without reasoning at the low-level state space $\mathcal{S}$, which is continuous, high-dimensional, and impractical for classical search. The goal of *Skill Model Learning* is to acquire a symbolic abstraction of skills that enables efficient composition via classical planning.

**Environment.** An *environment* is a tuple $(\mathcal{S}, \mathcal{T}, \Omega, T)$, where $\mathcal{S}$ is the continuous state space, $\mathcal{T}$ is a finite set of object types, and $\Omega$ is the skill library. Each skill $\omega \in \Omega$ is parameterized by object types drawn from the set $\mathcal{T}$. The environment dynamics are governed by an unknown transition function $T : \mathcal{S} \times \Omega \rightarrow \mathcal{S}$. A *setting* is defined as $(s_0, \mathcal{O})$, consisting of an initial state $s_0 \in \mathcal{S}$ and a set of typed objects $\mathcal{O}$ with $\tau(o) \subseteq \mathcal{T}$ for each $o \in \mathcal{O}$.

**Black-box skills.** A skill $\omega \in \Omega$ is a tuple $(\mathcal{I}_\omega, \pi_\omega, \beta_\omega, \theta_\omega)$, where $\mathcal{I}_\omega \subseteq \mathcal{S}$ is the *initiation set*, $\pi_\omega$ is the option policy, $\beta_\omega \subseteq \mathcal{S}$ is the *termination set*, and $\theta_\omega = (\tau_\omega^1, \ldots, \tau_\omega^k)$ are the type constraints

on its $k$ parameters. A skill instance $\underline{\omega} = \omega(o_1, \ldots, o_k)$ is valid for $o_i \in \mathcal{O}$ if $\tau_\omega^i \subseteq \tau(o_i)$ for all $i$. Executing $\omega$ from $s \in \mathcal{I}_\omega$ terminates in some $s' \in \beta_\omega$, affecting only the state of the bound objects.

**Symbolic predicates and operators.** To enable abstract reasoning, we introduce a finite set of predicates $\mathcal{P}$. Each $\sigma \in \mathcal{P}$ is parameterized by $\mathcal{T}$ and has a classifier $\phi_\sigma : \mathcal{O}^n \times \mathcal{S} \rightharpoonup \{0, 1\}$ that evaluates its truth value for arguments in a state. Grounding predicates with valid objects induces a set of grounded predicates $\bar{\mathcal{P}}$, and hence an abstract state space $\bar{\mathcal{S}} = 2^{\bar{\mathcal{P}}}$. The *abstraction function* $\Gamma : \mathcal{S} \to \bar{\mathcal{S}}$ maps each state to the set of grounded predicates true in that state, while the *grounding function* $\mathcal{G} : \bar{\mathcal{S}} \to \mathcal{S}$ maps an abstract state to the states in which its grounded predicates hold.

An operator $a \in \mathcal{A}$ is defined as $(\omega_a, \theta_a, \text{PRE}_a, \text{EFF}_a)$, where $\omega_a \in \Omega$ is the associated skill, $\theta_a$ are typed operator parameters, $\text{PRE}_a \subseteq \mathcal{P}$ is the precondition set, and $\text{EFF}_a = (\text{EFF}_a^+, \text{EFF}_a^-)$ are the add and delete effects. Grounding all operators $a \in \mathcal{A}$ with objects $o \in \mathcal{O}$ yields a set of abstract actions $\bar{a} \in \bar{\mathcal{A}}$, each executable whenever its ground preconditions hold in the current abstract state.

**Planning problems.** Given an environment $(\mathcal{S}, \mathcal{T}, \Omega, T)$, the agent's objective is to learn an abstract transition model $\mathcal{M} = (\mathcal{P}, \mathcal{A})$, such that $\mathcal{M}$ is sufficient for planning with $\Omega$. Formally, a *skill planning problem* is $(s_0, \mathcal{O}, \mathcal{S}_g)$, where $(s_0, \mathcal{O})$ is a setting and $\mathcal{S}_g \subseteq \mathcal{S}$ are goal states. The corresponding *abstract planning problem* is $(\Gamma(s_0), \bar{\mathcal{A}}, \bar{\mathcal{S}}_g)$, where $\bar{\mathcal{S}}_g = \{\bar{s} \in \bar{\mathcal{S}} \mid \mathcal{G}(\bar{s}) \cap \mathcal{S}_g \neq \emptyset\}$. An abstract plan $\bar{\pi} = \langle \bar{a}_1, \ldots, \bar{a}_n \rangle$ is *valid* if its execution under $\mathcal{M}$ yields an abstract trajectory consistent with some feasible low-level trajectory under $T$, ending in a goal state.

**Problem statement.** The *Skill Model Learning* problem is: given experience in the form of *state–skill-instance–next-state* tuples $\langle s, \underline{\omega}, s' \rangle$, learn a model $\mathcal{M} = (\mathcal{P}, \mathcal{A})$ such that every plan found by a complete symbolic planner over $\mathcal{M}$ corresponds to a feasible low-level skill plan.

## 3 SKILLWRAPPER

In this section, we introduce SKILLWRAPPER, a novel approach that autonomously learns symbolic representations for black-box skills using the concepts defined in Appendix C. To produce a valid abstract model that enables planning, SKILLWRAPPER iterates through a three-step process: *1)* actively proposing and executing exploratory skill sequences to collect data on the initiation and termination set of each skill, *2)* incrementally building a set of predicates from scratch by contrasting positive and negative examples, and then *3)* constructing valid operators using these invented predicates, from which further exploratory skill sequences can be proposed. This procedure is outlined in Algorithm 1. As SKILLWRAPPER continues to collect data, add predicates, and update its planning model, it learns a progressively more accurate abstract transition model. The resulting skill representations, or *operators*, can be used with an off-the-shelf classical planner to solve task planning problems specified using RGB images. We delve into the core components of our system (lines 4–9 of Algorithm 1) in the following subsections. Lastly, we provide strong theoretical results for the soundness and completeness of SKILLWRAPPER in Section 3.4.

---

**Algorithm 1** SKILLWRAPPER

1: **Input:** Set of skills $\Omega$, number of iterations $m \in \mathbb{N}_1$
2: **Output:** Abstract transition model $\mathcal{M} = (\mathcal{P}, \mathcal{A})$
3: $\mathcal{D}, \mathcal{P}, \mathcal{A} \leftarrow \emptyset$
4: **for** $i \in \{1, \ldots, m\}$ **do**
5:      $\langle \underline{\omega}_1, \ldots, \underline{\omega}_k \rangle \leftarrow$ PROPOSESKILLSEQUENCE$(\Omega, \mathcal{D}, \mathcal{P}, \mathcal{A})$
6:      $\mathcal{D} \leftarrow \mathcal{D} \, \| \, $EXECUTESKILLS$(\underline{\omega}_1, \ldots, \underline{\omega}_k)$
7:      $\mathcal{P} \leftarrow$ INVENTPREDICATES$(\Omega, \mathcal{D}, \mathcal{P}, \mathcal{A})$
8:      $\mathcal{A} \leftarrow$ LEARNOPERATORS$(\mathcal{D}, \mathcal{P})$
9: **end for**
10: **return** $\mathcal{M}$

---

### 3.1 ACTIVE DATA COLLECTION

To collect data for learning, our method commands the agent to execute its skills in the world and then collects the resulting transitions. Each command is a sequence of skill instances $\langle \underline{\omega}_1, \ldots, \underline{\omega}_k \rangle$.

While executing these commands, the agent collects a dataset $\mathcal{D}$ of transitions of the form $\langle s, \underline{\omega}, s' \rangle$, where $s, s' \in \mathcal{S}$. These transitions can answer two questions:

1. *Executability*: Can the skill instance $\underline{\omega}$ be executed from state $s$?

2. *Skill Dynamics*: If $s \neq s'$, what has changed in the environment due to executing $\underline{\omega}$?

We guide the exploration of skill preconditions and effects using a foundation model, which proposes skill sequences in natural language. Rather than naively sampling these sequences from the foundation model's token distribution, we prompt the foundation model for a batch of candidate skill sequences and apply two scoring functions to bias the system toward promising sequences that explore novel skill instance pairs and keep a balance between success and failure executions (see Appendix B.1 for details). SKILLWRAPPER facilitates the efficient data collection process, which results in a dataset $\mathcal{D}$ that is critical to downstream processes, such as guiding predicate invention with failure transitions, and improving the learned abstract model by eliminating unnecessary preconditions, etc.

## 3.2 Predicate Invention

We now present our predicate invention algorithm, which, unlike prior work (Silver et al., 2023; Liang et al., 2025), does not require an initial set of predicates to bootstrap the invention process.

**Conditions for predicate invention.** SKILLWRAPPER identifies two conditions under which the current predicate vocabulary is insufficient, based on the desired properties of soundness and completeness (discussed fully in Appendix C). In these cases, the system must invent new predicates to resolve incongruities between the observed data and the current abstract transition model. We illustrate how SKILLWRAPPER can achieve the desired model properties in the Venn diagrams in Appendix D.1.

To formally describe the conditions, we define two sets, $\boldsymbol{\alpha}_\omega$ and $\boldsymbol{\zeta}_\omega$, representing the states in which the model predicts that the skill may either be initiated (when $s \in \boldsymbol{\alpha}_\omega$) or terminated (when $s \in \boldsymbol{\zeta}_\omega$), respectively. Both sets are derived from and defined with the learned operators, and their formal definitions can be found in Appendix C.

The first condition arises when SKILLWRAPPER detects that the symbolic vocabulary cannot express a necessary precondition for a skill. Concretely, this occurs when two transitions involve instances of the same skill, one successful and one failed, both satisfy the initiation condition of the skill under the current predicates. Because the vocabulary cannot distinguish between these initial states, an additional predicate is required. Formally, this condition is expressed as:

$$\exists \langle s_i, \underline{\omega}_i, s'_i \rangle, \langle s_j, \underline{\omega}_j, s'_j \rangle \in \mathcal{D} \text{ s.t. } s_i \in \boldsymbol{\alpha}_{\underline{\omega}_i}, s_j \in \boldsymbol{\alpha}_{\underline{\omega}_j}, \text{ but } s_i \in \mathcal{I}_{\underline{\omega}_i} \text{ while } s_j \notin \mathcal{I}_{\underline{\omega}_j}.$$

The second condition used by SKILLWRAPPER to trigger predicate invention is based on inconsistencies in observed skill effects. Specifically, this occurs when two transitions that involve instances of the same skill produce identical abstract effects, despite one succeeding and the other failing. In a deterministic setting, this reduces to a successful skill execution producing no effects, though the condition naturally extends to stochastic settings with mid-execution failures. Formally, we express the condition as:

$$\exists \langle s_i, \underline{\omega}_i, s'_i \rangle, \langle s_j, \underline{\omega}_j, s'_j \rangle \in \mathcal{D} \text{ s.t. } s'_i \in \boldsymbol{\zeta}_{\underline{\omega}_i}, s'_j \in \boldsymbol{\zeta}_{\underline{\omega}_j}, \text{ but } s_i \in \mathcal{I}_{\underline{\omega}_i} \text{ and } s_j \notin \mathcal{I}_{\underline{\omega}_j}.$$

**Contrastive predicate proposal** When a satisfying transition pair is identified under the conditions of predicate invention, SKILLWRAPPER prompts the foundation model with the two transitions and their corresponding states (RGB images) to propose a candidate predicate that can potentially distinguish the transition pair. The transition pair offers contrastive clues for the foundation model to propose predicates that precisely resolve the incongruity.

Figure 2: **Example of Predicate Invention.** The initial states of two transitions are both said to satisfy the preconditions of certain operators learned from the same skill, while transition 1 is successful, but transition 2 is not. In this case, the first condition (precondition) is triggered, and the foundation model is prompted with both transitions to invent a new predicate.

**Empirical predicate selection.** Although foundation models provide a strong prior on which information is skill-relevant for predicate construction, they may still produce errors and hallucinations. To ensure robustness, we introduce a scoring function that estimates the usefulness of a candidate predicate by adding it to the vocabulary and learning hypothetical operators. For each successful transition in $\mathcal{D}$, there must exist at least one operator with preconditions satisfied by the transition's initial state; for each failed transition, no such operator may exist. Effect evaluation follows the same principle. After evaluating all transitions, we decide whether to add the candidate predicate based on an empirical threshold (Details of Algorithm 6 can be found in the appendix.)

### 3.3 OPERATOR LEARNING

Our operator learning procedure extends the *associative model learning* paradigm (Arora et al., 2018) to the setting of skill abstraction.

**Associative model learning.** A single skill may induce multiple distinct abstract state changes depending on the context of execution. To represent these *conditional effects*, SKILLWRAPPER clusters observed transitions based on their lifted (object-agnostic) effect sets, enabling it to learn a single operator across distinct instantiations of a skill. The preconditions of each operator are then computed as the intersection of all initial abstract states in the corresponding transitions, ensuring that each operator is both minimal and consistent.

**Multi-type object-centrism.** In realistic domains, objects do not fit neatly into single type categories, but rather belong to multiple overlapping categories (e.g., a Cup is `fillable`, while a `Bottle` is both `fillable` and `openable`). This complicates the process of generalizing grounded transitions into lifted operators, because it may be ambiguous which object attribute enables successful execution. We adopt a conservative strategy: SKILLWRAPPER assigns arguments of each operator using the lowest level of the type hierarchy consistent with the data, preventing over-generalization while retaining compositional structure.

**Predicate re-evaluation.** Predicates are generated sequentially, and early inventions may bias later stages if left unchecked. To mitigate this, SKILLWRAPPER re-applies the scoring function to the entire predicate set after each iteration of data collection. This allows spurious or redundant predicates to be discarded as more data is collected. In addition, tautological predicates—those that are always true or always false—are automatically filtered. As a result, the learned predicate set remains compact, informative, and aligned with the most recent transition data.

### 3.4 THEORETICAL ANALYSIS

We now provide theoretical guarantees for SKILLWRAPPER. These results show that the learned symbolic model is sound with respect to observed data and converges to a complete model with high probability. Full proofs are deferred to Appendix D.

**Theorem 1** (Soundness of SKILLWRAPPER). *Every operator $a \in \mathcal{A}_n$ in the model $\mathcal{M}_n$ learned by SKILLWRAPPER is supported by at least one observed transition $\langle s, \omega, s' \rangle \in B_n$. That is, $\underline{s} \models \text{PRE}_a$ and $\underline{s}' \models \text{EFF}_a$.*

*Sketch.* SKILLWRAPPER constructs each operator directly from sampled transitions; thus no unsupported operator can appear.

**Lemma 1.** *For each $\omega \in \Omega$, the initiation set $I_\omega$ and termination set $\beta_\omega$ inferred by* SKILLWRAPPER *match exactly the corresponding predicate sets $\boldsymbol{\alpha}_\omega$ and $\boldsymbol{\zeta}_\omega$ derived from $B_n$, i.e. $I_\omega = \boldsymbol{\alpha}_\omega$ and $\beta_\omega = \boldsymbol{\zeta}_\omega$.*

*Sketch.* Any mismatch would contradict the termination condition of SKILLWRAPPER; thus initiation and termination sets are consistent with observed data.

**Theorem 2** (Probabilistic-completeness of SKILLWRAPPER). *Let $\mathcal{M}^*$ be the true complete model. With probability at least $1 - |\mathcal{H}| \exp(-n\epsilon)$, the model $\widehat{\mathcal{M}}_n$ learned from $n$ i.i.d. samples satisfies $d_{\mathrm{compl}}(\widehat{\mathcal{M}}_n, \mathcal{M}^*) \leq \epsilon$, i.e. it misses fewer than an $\epsilon$-fraction of feasible transitions.*

*Sketch.* Since $\widehat{\mathrm{Err}}(\widehat{\mathcal{M}}_n) = 0$ by construction, a Chernoff bound and union bound over the finite hypothesis class imply that the true error is small with high probability.

Together, these results establish that SKILLWRAPPER learns symbolic operators that are sound with respect to observed transitions, consistent in their preconditions and effects, and probabilistically complete relative to the true underlying model. These properties justify SKILLWRAPPER as a reliable model-learning procedure for planning. Next, we discuss empirical evaluation for SKILLWRAPPER.

## 4 EXPERIMENTS

For all experiments in this section, we consider images as fully observable state representations, assuming that an abstract state can be inferred from an image without uncertainty. Both the initial and goal states of each problem are specified using RGB images. These images may come from diverse sources, including a top-down view of an animated game, a third-person camera observing a robot, or the robot's own egocentric perspective. All quantitative results reported in this section are averaged over five independent runs for simulation experiments and three runs for real robot experiments.

### 4.1 IMPLEMENTATION OF PREDICATES

We employ foundation models (specifically vision-language models or VLMs) for both predicate generation and evaluation:

- A foundation model gives us a string that can be used as a lifted predicate (generates interpretable relational classifiers with a good heuristic).
- After grounding with valid parameters, the predicate can be prompted to the foundation model again to acquire the truth value. In other words, a foundation model can be used as a relational classifier.

With these two properties, we can use the VLM's response in string form as a relational predicate, and a grounded version of the predicate can be used as a classifier. We use GPT-5 (OpenAI, 2025) for predicate generation and evaluation. In addition to the system performance reported in the main paper, we also conducted comprehensive studies of the component-wise reliability of the VLM in Appendix G.

### 4.2 SIMULATION

We first conduct experiments in Robotouille (Gonzalez-Pumariega et al., 2025), which is a simulated grid world kitchen domain with an agent that has five high-level skills: *Pick*, *Place*, *Cut*, *Cook*, and *Stack*. In the environment, there are several objects: a patty, lettuce, a top bun, and a bottom bun; there is also a cutting board and a stove for cutting the lettuce and cooking the patty, respectively. We design and categorize 50 abstract planning problems: 20 easy problems, whose optimal solutions have no more than 7 steps; 20 hard problems, whose optimal solutions have no more than 15 steps; and 10 impossible tasks that cannot be realized in the environment.

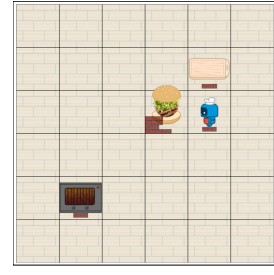

Figure 3: Screenshot of the Robotouille environment.

For a complete evaluation of SKILLWRAPPER, we compare SKILLWRAPPER against four baselines:

- **Expert Operators**: A human expert who is familiar with PDDL is asked to interact with the environment and manually write predicates with semantics and PDDL operators.

- **System Predicates**: This baseline directly uses the built-in predicate set of the simulator, which is designed to define any possible simulated state unambiguously. In addition, instead of getting the truth values through classification with foundation models, this baseline has access to the exact abstract states of the simulator. However, predicate invention is disabled in this environment.

- **ViLa** (Hu et al., 2023): This baseline is a closed-loop VLM-based approach that iteratively prompts a foundation model for the next action until the goal state is reached, given an image observation and the agent's action history.

- **Random Exploration**: Instead of proposing skill sequences, this baseline randomly samples a skill and populates the arguments with valid objects. This baseline shares the same predicate invention and operator learning algorithms as SKILLWRAPPER.

- **No Heuristic**: This baseline is the same as SKILLWRAPPER, except that skill sequences are selected randomly from the foundation model's output without applying the heuristics.

For each baseline that performs operator learning, we run the learning algorithm for five iterations, with each iteration proposing and executing one skill sequence consisting of 15 steps as their interaction budgets. We then evaluate each method on the evaluation set and report the average results in Table 1, where *Solved %* is the percentage of the problem set that was successfully solved or where impossible tasks were correctly identified by returning an empty plan, and *PB* stands for planning budgets—the number of plans that were tried before solving the problem (adopted from Liang et al. (2025)). We set a planning budget cap of 10 across all problems; if the planning budget has been used up for a problem, it is considered a failure. Theoretically, *PB* is an adequate metric that reflects the completeness of the learned model, and the *impossible* problems reflect its soundness.

As shown in the table, SKILLWRAPPER outperforms all baselines that have no access to privileged knowledge, and even surpasses the performance of the system predicates baseline (Sys Preds.) on hard problems. Here we present the key insights, while leaving case studies, example operators, and failure modes and analysis of SKILLWRAPPER in Appendix F.

Table 1: Baseline Comparison in Robotouille Environment

| Method | Easy | | Hard | | Impossible |
|---|---|---|---|---|---|
| | Solved % ↑ | PB ↓ | Solved % ↑ | PB ↓ | |
| Expert Ops. | **81.0 ± 3.7** | **1.9 ± 0.4** | **58.1 ± 3.9** | **4.2 ± 0.4** | **100 ± 0.0** |
| Sys Preds. | 79.0 ± 3.7 | 2.6 ± 0.2 | 22.0 ± 12.9 | 7.8 ± 1.3 | 42.0 ± 7.5 |
| ViLa | 46.0 ± 16.2 | - | 13.9 ± 11.6 | - | 20.0 ± 10.9 |
| Random Exp. | 4.0 ± 2.0 | 9.6 ± 0.2 | 0 ± 0.0 | 10.0 ± 0.0 | 100 ± 0.0 |
| No Heuristic | **76.0 ± 4.9** | **2.5 ± 0.9** | 24.0 ± 19.6 | 7.8 ± 1.8 | 80 ± 20.9 |
| **Ours** | 74.0 ± 3.7 | 2.7 ± 0.4 | **40.0 ± 3.2** | **6.3 ± 0.4** | **100 ± 0.0** |

Compared to the expert-constructed operators, which serve as an approximate upper bound for performance without privileged simulator access, SKILLWRAPPER demonstrates competitive accuracy while requiring only a small number of exploratory interactions. In particular, SKILLWRAPPER exhibits strong generalization from the easy to the hard set, indicating that the invented predicates and learned operators capture meaningful abstractions rather than overfitting to specific training traces. The gap between SKILLWRAPPER and ViLa highlights the benefit of explicitly learning a symbolic model instead of relying solely on open-loop prompting, while the poor performance of Random Exploration underscores the importance of guided data collection in learning effective operators, and the unstable performance of No Heuristic in hard problems shows the importance of a sufficient predicate set and how SKILLWRAPPER manages to reliably explore the state space and learn it. These findings suggest that SKILLWRAPPER achieves a favorable trade-off between data efficiency and model soundness and completeness, which is crucial for scaling to larger domains.

## 4.3 REAL ROBOTS

To demonstrate the applicability of SKILLWRAPPER for real-world agentic settings, we designed two sets of experiments with two robotic platforms: a Franka Emika Panda robot (Figures 4a and 4b) and a bimanual platform with two Kuka iiwa robots (Figures 4c and 4d). For both robot experiments, we assume that all skills are deterministic, which is a common assumption made by existing work (Silver et al., 2023; Han et al., 2024; Liang et al., 2025; Athalye et al., 2025). We give SKILLWRAPPER 15 steps as its interaction budget per iteration. The supplementary material contains video demonstrations of both experiments.

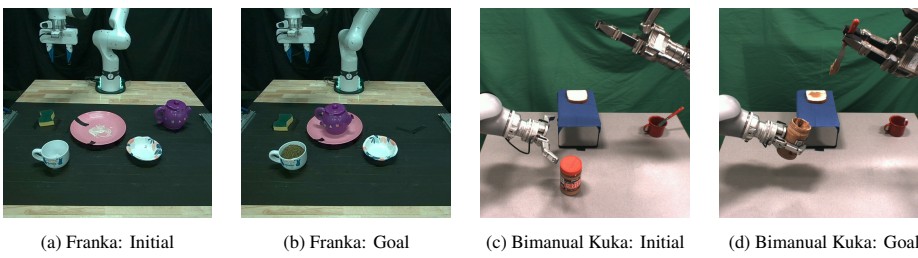

(a) Franka: Initial       (b) Franka: Goal       (c) Bimanual Kuka: Initial       (d) Bimanual Kuka: Goal

Figure 4: Initial and Goal States for Real Robot Experiments.

**Generalization of** SKILLWRAPPER. In this setting, a tabletop Franka Emika Research 3 (Panda) manipulator has its skill set $\Omega$ consisting of five black-box skills: *Pick*, *Place*, *Stack*, *Pour*, and *Wipe*. The object set $\mathcal{O}$ contains five objects: a mug, a teapot, a plate, a bowl, and a sponge. The robotic agent can pick and place all objects except the mug and plate, pour ingredients from the teapot into the mug, and use the sponge to wipe the plate if it is dirty. To validate the generalization ability of the learned skill representations, we design three smaller training environments, such that each environment only contains a subset of $\mathcal{O}$, and thus only a subset of $\Omega$ are executable. Each of the training environments contains fewer than 10 possible states. After running SKILLWRAPPER for exactly one iteration for each environment, we port the learned skill representations to the test environment that contains all objects in $\mathcal{O}$, which induces a state space of 34 possible abstract states. To quantify this generalization process, we similarly prepare an evaluation problem set that consists of five problems in the training environments, five problems in the test, and five *Impossible* problems across the two environments. The results are shown in Table 2.

Table 2: Results of Generalization Experiment

| Method | In-domain | | Generalization | | Impossible |
|---|---|---|---|---|---|
| | Solved % ↑ | PB ↓ | Solved % ↑ | PB ↓ | |
| Expert Ops. | 66.7 ± 9.4 | 3.3 ± 0.9 | 53.3 ± 9.4 | 5.3 ± 0.9 | 46.7 ± 0.0 |
| ViLa | 46.7 ± 9.4 | - | 6.7 ± 9.4 | - | 6.7 ± 9.4 |
| **Ours** | **76.7 ± 9.4** | **2.7 ± 0.9** | **60.0 ± 0.0** | **4.0 ± 0.0** | **66.7 ± 9.4** |

**Learning Curve of** SKILLWRAPPER. In this bimanual manipulation setting, a robot with two Kuka iiwa arms is equipped with a skill set $\Omega$ containing six black-box skills: *LeftArmPick*, *RightArmPick*, *Open*, *Scoop*, *Spread*, and *Drop*. The object set $\mathcal{O}$ consists of three objects: a peanut butter jar, a knife, and a slice of bread. This robot can pick up the knife and jar, drop the knife, open the jar, scoop peanut butter with the knife, and spread it on the bread. Notably, this environment contains multiple dead ends, which would hinder the data gathering process. For example, the knife cannot be picked up again once dropped, the jar cannot be released once picked up, and the bread and knife cannot be cleaned once in contact with peanut butter. Moreover, the skills are heavily interdependent. We designed the experiment in this way to investigate the learning process of SKILLWRAPPER over several iterations. Again, we compare the performance against two baseline methods, ViLa and Expert Operators. An example of predictive truth value changes induced by a sequence is shown as in Figure 5. We observe that performance improves as SKILLWRAPPER progressively obtains more transition data and invents more predicates, finally surpassing the baselines. The performance improvement over iterations is shown in Figure 6.

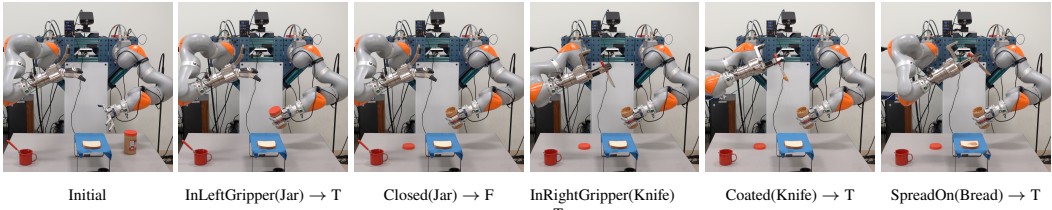

| Initial | InLeftGripper(Jar) → T | Closed(Jar) → F | InRightGripper(Knife) → T | Coated(Knife) → T | SpreadOn(Bread) → T |

Figure 5: Sequence of Bimanual Robot Skill Execution with Predicate Value Changes

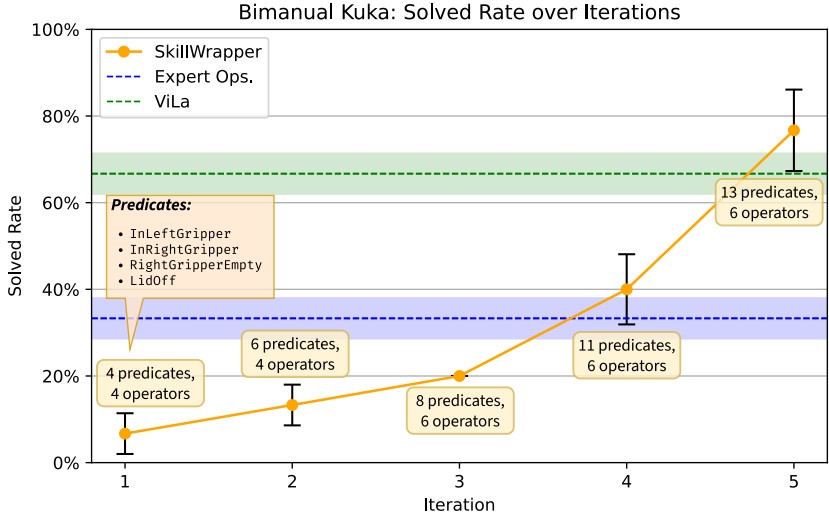

Figure 6: Bimanual Kuka Scenario Results over 5 iterations with invented predicate and learned operator count. As the number of predicates and operators grows, SKILLWRAPPER improves over the baseline methods.

**Discussion.** Our results demonstrate that SKILLWRAPPER is effective in real robot settings: our method generalizes skill representations learned in restricted domains to richer environments and progressively improves in more challenging scenarios with irreversible actions and interdependent skills. By outperforming both expert-defined operators and baseline methods, SKILLWRAPPER highlights the importance of predicate invention and iterative learning for scaling symbolic representations to embodied tasks.

## 5 RELATED WORKS

Our work uses pre-trained foundation models for learning symbolic representations of black-box robot skills useful to planning and close to human language and understanding. This work draws ideas from different fields of research such as model learning, abstraction learning, and task and motion planning (TAMP). Several methods have used foundation models (mainly LLMs) as high-level planners (Ahn et al., 2022; Rana et al., 2023; Driess et al., 2023). Several approaches have used foundation models as robot action models (Brohan et al., 2023; Shridhar et al., 2023) or to generate reward functions for robot tasks (Wang et al., 2024b). Concurrent work has also explored how representations can be learned directly from pixels (Athalye et al., 2025). Although these approaches show promising results for short-horizon single-skill problems, they fail to scale to complex long-horizon problems (Kambhampati et al., 2024). Lastly, multiple approaches (Han et al., 2024; Liang et al., 2025) have used foundation models to learn symbolic representations of robot skills, but require extensive feedback or prior knowledge from human experts. To the best of our knowledge, our work is the first to use a foundation model to automatically learn the human-interpretable symbolic characterization of robot skills with *theoretical guarantees*.

TAMP has long been used to solve complex robot tasks (Dantam et al., 2018; Shah et al., 2020; Garrett et al., 2021). However, these approaches require symbolic models of the robot skills for task planning. Various approaches have been developed to learn such symbolic models compatible with TAMP solvers from high-dimensional inputs (Konidaris et al., 2018; Silver et al., 2023; Shah et al., 2024). Additionally, the abstract representations learned through these methods are not human-interpretable. On the other hand, we explicitly design our approach to work with high-dimensional inputs and generate human-interpretable abstractions using pre-trained foundation models. We consider abstractions as human-interpretable if they are semantically meaningful and use informative language descriptions. SKILLWRAPPER also connects to other domains in robotics, and the full related work can be found in Appendix I.

## 6 CONCLUSION

We characterize important properties of a learned symbolic model and present the first known approach that employs off-the-shelf foundation models to invent symbolic representations for black-box skills of an agent while providing strong guarantees of soundness and completeness of the learned representations. By combining these theoretical guarantees with foundation model-driven data collection and predicate evaluation, SKILLWRAPPER produces interpretable operators directly usable by classical planners. Empirical results in a simulated burger domain and on real robots demonstrate that SKILLWRAPPER enables efficient long-horizon planning without hand-engineered abstractions, offering a principled path towards scalable skill reasoning.

## 7 REPRODUCIBILITY STATEMENT

To ensure reproducibility of our work, we provide source code and the prompts used in our experiments as supplementary materials. Although the reproducibility of real-world robot experiments is limited by hardware, simulation experiments run in Robotouille should be reliably reproduced, granted that the checkpoints of the foundation model (i.e., OpenAI's GPT-5 (OpenAI, 2025)) have not been moved.

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

APPENDIX

# A    FORMAL FRAMEWORK

We consider a problem setting in which an agent is equipped with a set of predefined, "black-box" skills. The agent can evaluate whether a skill is executable in the current state, but it does not have a complete transition model of the skills a priori, and therefore cannot compose its skills to solve long-horizon problems without considering low-level details. However, if the agent were to learn symbolic models of its skills, it could use classical planning to efficiently compose them to solve new tasks. In this section, we formalize this setting as the problem of *Skill Model Learning*.

## A.1    PRELIMINARIES

**Environment Model.**    We define an *environment* as a tuple $(\mathcal{S}, \mathcal{T}, \Omega, T)$, where the *state space* $\mathcal{S}$ is assumed to be high-dimensional, continuous, and fully observable; the *type set* $\mathcal{T}$ enumerates the possible object types; and the *skills* $\Omega$ are object-centric, such that each $\omega \in \Omega$ is parameterized by object types drawn from the *type set* $\mathcal{T}$. The *transition function* $T : \mathcal{S} \times \Omega \to \mathcal{S}$ characterizes the environment dynamics but is unknown to the agent. For example, Pour(?teapot, ?mug) may be used to pour tea from a teapot into a mug. However, certain environmental aspects (e.g., which mugs are available to pour into) may differ between settings during learning. We therefore define a specific *setting* by the tuple $(s_0, \mathcal{O})$, specifying an initial state $s_0 \in \mathcal{S}$ and a set of typed objects $\mathcal{O}$, where the type(s) of object $o \in \mathcal{O}$ are denoted by $\mathtt{t}_o \subseteq \mathcal{T}$.

**Black-box Skills.**    We model skills as object-centric options (Sutton et al., 1999) with discrete, object-typed parameters. Formally, a *skill* $\omega \in \Omega$ is defined by a tuple $(\mathcal{I}_\omega, \pi_\omega, \beta_\omega, \Theta_\omega)$, where the *initiation set* $\mathcal{I}_\omega \subseteq \mathcal{S}$ contains states from which the skill may be executed; the *policy* $\pi_\omega$ controls the agent during the skill; the *termination set* $\beta_\omega \subseteq \mathcal{S}$ is the set of states at which the skill immediately terminates; and the *skill parameters* $\Theta_\omega = (\theta_\omega^1, \dots, \theta_\omega^k)$ specify type constraints $\mathtt{t}_{\theta_\omega^i} \subseteq \mathcal{T}$ for valid skill arguments. Specifically, a skill $\omega \in \Omega$ may be instantiated to create a *skill instance* $\underline{\omega}$ using objects $o_{1:k} \in \mathcal{O}^k$ if and only if for $1 \leq i \leq k$, $\mathtt{t}_{\theta_\omega^i} \subseteq \mathtt{t}_{o_i}$. We assume that executing a skill only affects the state of objects passed as arguments.[1]

**Symbolic Abstractions.**    Because the agent must evaluate $\mathcal{I}_\omega$ on individual states, it cannot distinguish essential skill information from irrelevant details, making long-horizon planning combinatorially difficult. However, each skill affects only a few objects at once, leaving the rest of the world unchanged. The agent can exploit this property by instead learning a factored state representation—formalized here as first-order logic models in PDDL (McDermott et al., 1998)—providing an abstract transition model of its skills. Such abstractions enable the agent to use classical planning to compose its skills and accomplish unseen, long-horizon goals.

We use symbolic *predicates* $\mathcal{P}$ to express abstract relations between objects. Each predicate $\sigma \in \mathcal{P}$ is a tuple $(C_\sigma, \Theta_\sigma)$, where the *predicate classifier* $C_\sigma$ tests whether the predicate holds in a state, given a binding of objects to the *predicate parameters* $\Theta_\sigma = (\theta_\sigma^1, \dots, \theta_\sigma^n)$. Each parameter $\theta_\sigma^i$ specifies a type constraint $\mathtt{t}_{\theta_\sigma^i} \subseteq \mathcal{T}$ on corresponding object arguments. Formally, the partial function[2] $C_\sigma : \mathcal{O}^n \rightharpoonup (\mathcal{S} \to \{0, 1\})$ is defined on objects $o_{1:n} \in \mathcal{O}^n$ if and only if $\mathtt{t}_{\theta_\sigma^i} \subseteq \mathtt{t}_{o_i}$ for $1 \leq i \leq n$. In such cases, we say that the objects $o_{1:n}$ are *valid arguments* for $\sigma$.

By observing different effects, the agent must construct a set of *operators* $\mathcal{A}$ that define abstract transition models for the agent's skills. We define each operator $a \in \mathcal{A}$ by $(\omega_a, \Theta_a, \text{PRE}, \text{EFF}^+, \text{EFF}^-)$, where $\omega_a \in \Omega$ is the corresponding skill; the *operator parameters* $\Theta_a = (\theta_a^1, \dots, \theta_a^m)$ impose type constraints $\mathtt{t}_{\theta_a^i} \subseteq \mathcal{T}$ on operator arguments; the *preconditions* PRE are a conjunction of literals over $\mathcal{P}$ defining the conditions necessary to apply the operator; and the *add* and *delete effects*, $\text{EFF}^+$ and $\text{EFF}^-$, are the subsets of $\mathcal{P}$ that become true and false, respectively, after the operator is applied.

**Grounded Abstractions.**    To apply an abstract transition model to a real-world setting, the agent must map the low-level state to an abstract state, which requires *grounding* the known predicates

---

[1]We do not, however, assume that a skill necessarily affects the state of *all* objects passed in as arguments.
[2]We denote a partial function using $f : \mathcal{X} \rightharpoonup \mathcal{Y}$.

using concrete objects so that their truth value may be determined. A predicate $\sigma \in \mathcal{P}$ may only be grounded using valid arguments, inducing a *grounded predicate* $\underline{\sigma} = \sigma(o_{1:n})$. Given a low-level state, the classifier $C_{\underline{\sigma}} : \mathcal{S} \to \{0, 1\}$ tests whether the predicate holds for the object arguments. We define the set of *grounded predicates* for a setting $(s_0, \mathcal{O})$ under predicates $\mathcal{P}$ as $\underline{\mathcal{P}} = \{\sigma(o_{1:n}) : \sigma \in \mathcal{P}, o_{1:n} \in \mathcal{O}^n, \bigwedge_{i=1}^{n} \mathsf{t}_{\theta_{\sigma}^i} \sqsubseteq \mathsf{t}_{o_i}\}$, inducing an *abstract state space* $\underline{\mathcal{S}} = 2^{\underline{\mathcal{P}}}$ where each abstract state corresponds to a specific combination of grounded relations.

Given a set of operators $\mathcal{A}$ in a setting $(s_0, \mathcal{O})$, the *abstract action space* $\underline{\mathcal{A}}$ is the set of all valid groundings of the operators using objects from $\mathcal{O}$. Each *abstract action* (i.e., grounded operator) is defined as $\underline{a} = a(o_{1:m})$, where $a \in \mathcal{A}$, $o_{1:m} \in \mathcal{O}^m$, and for $1 \leq i \leq m$, $\mathsf{t}_{\theta_a^i} \sqsubseteq \mathsf{t}_{o_i}$. Grounding an operator induces *ground preconditions* $\underline{\text{PRE}}$, which are a conjunction of ground literals over $\underline{\mathcal{P}}$; *ground add effects* $\underline{\text{EFF}}^+ \subseteq \underline{\mathcal{P}}$; and *ground delete effects* $\underline{\text{EFF}}^- \subseteq \underline{\mathcal{P}}$.

The *grounding function* $\mathcal{G} : \underline{\mathcal{P}} \to 2^{\mathcal{S}}$ maps each grounded predicate to its *grounding set* $\mathcal{G}(\underline{\sigma}) \subseteq \mathcal{S}$, defined as $\mathcal{G}(\underline{\sigma}) = \{s \in \mathcal{S} : C_{\underline{\sigma}}(s) = 1\}$. We overload this notation for abstract states $\underline{s} \in \underline{\mathcal{S}}$ so that $\mathcal{G}(\underline{s}) = \bigcap_{\underline{\sigma}_i \in \underline{s}} \mathcal{G}(\underline{\sigma}_i)$. Conversely, the *abstraction function* $\text{ABSTRACT} : \mathcal{S} \to \underline{\mathcal{S}}$ maps each low-level state $s \in \mathcal{S}$ to the abstract state $\underline{s} \in \underline{\mathcal{S}}$ defined by $\text{ABSTRACT}(s) = \{\underline{\sigma} \in \underline{\mathcal{P}} : C_{\underline{\sigma}}(s) = 1\}$.

## A.2 PROBLEM DEFINITION

**Definition 1.** *Given an environment* $(\mathcal{S}, \mathcal{T}, \Omega, T)$ *containing settings* $\{(s_0, \mathcal{O})\}_{i=1}^{N}$, *we define a* Skill Model Learning *problem as learning an* abstract transition model $\mathcal{M} = (\mathcal{P}, \mathcal{A})$ *for the skills* $\Omega$.

After a period of continual learning in one or more settings, an agent may be evaluated on an *skill planning problem* $\mathbf{p} = (s_0, \mathcal{O}, \mathcal{S}_g)$, where $\mathcal{S}_g \subseteq \mathcal{S}$ is the set of *goal states* to be reached. Given a model $\mathcal{M} = (\mathcal{P}, \mathcal{A})$, a classical planner can be used to search for an *abstract plan* $[\underline{a}_1, \ldots, \underline{a}_n]$ that solves the *abstract planning problem* $\underline{\mathbf{p}} = (\underline{s}_0, \mathcal{O}, \underline{\mathcal{S}}_g)$.

**Definition 2.** *An abstract plan* $[\underline{a}_1, \ldots, \underline{a}_n]$ *is called a* solution *for skill planning problem* $\mathbf{p} = (s_0, \mathcal{O}, \mathcal{S}_g)$ *iff for* $1 \leq i \leq n$, $s_i = T(s_{i-1}, \underline{\omega}_i), s_{i-1} \in \mathcal{I}_{\underline{\omega}_i}$, *and* $s_n \in \mathcal{S}_g$, *where* $\underline{\omega}_i = \underline{\omega}_{a_i}$.

In Sec. 3, we describe how SKILLWRAPPER constructs $\mathcal{M}$ from raw skill executions.

## B ALGORITHMS

### B.1 SKILL SEQUENCE PROPOSAL

Each skill sequence $\sigma = [\underline{\omega}_1, \ldots, \underline{\omega}_m]$ (Section 3.1) proposed by the foundation model is scored using two heuristics: coverage ($C$) and chainability ($Ch$). This section provides more details on how these heuristics are computed and algorithmically used to assign scores to each sequence.

**Overview.** We prompt a foundation model is provided with the agent's skill set $\Omega$ and the current set of abstract predicates $\mathcal{P}$ to generate and propose sequences of skills $\sigma$, with which the agent collects a dataset of transitions $\mathcal{D}$. The skill sequence proposal procedure (Algorithm 2) assigns a score tuple $(C, Ch)$ to all sequences and maintains a subset of pareto-front sequences that cannot strictly dominate another sequence, i.e., $(C_i < C_j) \vee (Ch_i < Ch_j) \vee (C_i \leq C_j \wedge Ch_i \leq Ch_j)$ where $i \neq j$. An output skill sequence is finally chosen from this pareto-front subset.

**Coverage** ($C$). Coverage (Algorithm 3) evaluates the information gain on all possible pairs of consecutively executed skills over existing transitions after executing a new skill sequence. Specifically, the information gain is measured by the increase in Shannon entropy (Shannon, 1948) over the distribution of all consecutive skill pairs resulting from executing the proposed skill sequence

$$C = \frac{\mathcal{Q}'}{\Sigma \mathcal{Q}'} \times log(\frac{\mathcal{Q}'}{\Sigma \mathcal{Q}'}) - \frac{\mathcal{Q}}{\Sigma \mathcal{Q}} \times log(\frac{\mathcal{Q}}{\Sigma \mathcal{Q}}) \tag{1}$$

where $\mathcal{Q}$ and $\mathcal{Q}'$ are matrices that tabulate the number of pairs of consecutively executed skills that occur before and after executing a new skill sequence, respectively. Maximizing coverage encourages the generation of proposed skill sequences that contain the least explored skill pairs. More importantly, this would allow our method to uncover a larger set of interdependencies across the preconditions and effects of all skills.

**Chainability** ($Ch$). Chainability predicts the ratio of successful to failed pairs of consecutively executed skills. By computing chainability, we estimate the degree to which the preconditions of operators learned in each iteration are satisfied, and executability can be inferred from the estimated symbolic states and the operators. With an appropriate chainability score, the collected dataset of skill execution traces maintains a balance between number of successful executions and failure executions, which is ideal for identifying possible mismatched pairs and thus inventing predicates.

---

**Algorithm 2** Propose Skill Sequences

---

1: **Input:** Skill set $\Omega$, skill execution traces $\mathcal{D}$, predicate set $\mathcal{P}$, operator set $\mathcal{A}$, batch size $n$
2: **Output:** Proposed skill sequence $\sigma$
3: seq_batch $\leftarrow$ GENERATESKILLSEQUENCES$(\Omega, n)$ ▷ Propose a batch of skill sequences with FM
4: Scores $\leftarrow \{\}$
5: **for** $\langle \underline{\omega}_1, \ldots, \underline{\omega}_k \rangle$ **in** seq_batch **do**
6:     cov $\leftarrow$ COVERAGE$(\mathcal{D}, \langle \underline{\omega}_1, \ldots, \underline{\omega}_k \rangle)$
7:     chain $\leftarrow$ CHAINABILITY$(\mathcal{A}, \langle \underline{\omega}_1, \ldots, \underline{\omega}_k \rangle)$
8:     Scores$[\langle \underline{\omega}_1, \ldots, \underline{\omega}_k \rangle] \leftarrow$ (cov, chain)
9: **end for**
10: **return** PARETOOPTIMAL(Scores)

---

**Algorithm 3** Coverage

---

1: **Input:** Skill execution traces $\mathcal{D}$, proposed skill sequence $\langle \underline{\omega}_1, \ldots, \underline{\omega}_k \rangle$
2: **Output:** Coverage score $C$
3: $\mathcal{Q} \leftarrow$ zero matrix of size $|\Omega| \times |\Omega|$ ▷ Construct a matrix of skill-pair counts
4: **for** $\langle s_i, \underline{\omega}_i, s_i' \rangle, \langle s_{i+1}, \underline{\omega}_{i+1}, s_{i+1}' \rangle$ in $\mathcal{D}$ **do** ▷ Iterate over all consecutive pairs of transitions
5:     $\mathcal{Q}[\underline{\omega}_i, \underline{\omega}_{i+1}] = \mathcal{Q}[\underline{\omega}_i, \underline{\omega}_{i+1}] + 1$
6: **end for**
7: $\mathcal{Q}' \leftarrow \mathcal{Q}$ ▷ New skill-pair count initialized
8: **for** $\langle \underline{\omega}_i, \underline{\omega}_{i+1} \rangle$ in $\langle \underline{\omega}_1, \ldots, \underline{\omega}_k \rangle$ **do**
9:     $\mathcal{Q}'[\underline{\omega}_i, \underline{\omega}_{i+1}] = \mathcal{Q}'[\underline{\omega}_i, \underline{\omega}_{i+1}] + 1$
10: **end for**
11: cov $\leftarrow$ COVERAGE$(\mathcal{Q}')$ $-$ COVERAGE$(\mathcal{Q})$ ▷ Compute coverage score using Eq. 1
12: **return** cov

---

**Algorithm 4** Chainability

---

1: **Input:** Operator set $\mathcal{A}$, Proposed skill sequence $\langle \underline{\omega}_1, \ldots, \underline{\omega}_k \rangle$
2: **Output:** Chainability score $chain$
3: exec_count $\leftarrow 0$ ▷ Total number of executable skills
4: sequence_length $\leftarrow$ LENGTH$(\langle \underline{\omega}_1, \ldots, \underline{\omega}_k \rangle)$
5: seq $\leftarrow [s_0]$ ▷ Store the trace of after-execution state
6: **for** $\underline{\omega}_i$ in $\langle \underline{\omega}_1, \ldots, \underline{\omega}_k \rangle$ **do**
7:     **for** $a \in \mathcal{A}_{\omega_i}$ **do**
8:         **if** $\Gamma(\text{seq}[-1]) \models \underline{\text{PRE}}_a$ **then** ▷ Successful execution predicated by the current model
9:             exec_count = exec_count + 1
10:             **break**
11:         **end if**
12:     **end for**
    $\underline{s}_{new} \leftarrow$ APPLYOPERATOR(seq$[-1].a$) ▷ Calculate the abstract state after execution
13:     seq $\leftarrow$ seq $\| \mathcal{G}(\underline{s}_{new})$ ▷ Append current low-level state to the trace
14: **end for**
15: chain $\leftarrow |\text{exec\_count}/\text{sequence\_length} - 0.5|$
16: **return** chain

---

## B.2 PREDICATE INVENTION

---

**Algorithm 5** Invent Predicates

---

1: **Input:** Skill set $\Omega$, skill execution traces $\mathcal{D}(\omega) = \{\langle s, \underline{\omega}, s' \rangle\}_\omega$, predicate set $\mathcal{P}$, operator set $\mathcal{A}$.
2: **Output:** Predicate set $\mathcal{P}$
3: **for** $\omega \in \Omega$ **do**
4:     **while** $\exists \langle s_i, \underline{\omega}_i, s_i' \rangle, \langle s_j, \underline{\omega}_j, s_j' \rangle \in \mathcal{D}$ s.t. $s_i \in \boldsymbol{\alpha}_{\underline{\omega}_i}, s_j \in \boldsymbol{\alpha}_{\underline{\omega}_j}$, but $s_i \in \mathcal{I}_{\underline{\omega}_i}, s_j \notin \mathcal{I}_{\underline{\omega}_j}$ **do**
5:         $\sigma \leftarrow$ NEWPREDICATE
6:         $\mathcal{P} \leftarrow \mathcal{P} \,\|\, \sigma$ **if** SCOREPRECOND$(\sigma, \mathcal{P}, \omega, \mathcal{D})$
7:     **end while**
8:     **while** $\exists \langle s_i, \underline{\omega}_i, s_i' \rangle, \langle s_j, \underline{\omega}_j, s_j' \rangle \in \mathcal{D}$ s.t. $s_i' \in \boldsymbol{\zeta}_{\underline{\omega}_i}, s_j' \in \boldsymbol{\zeta}_{\underline{\omega}_j}$, but $s_i \in \mathcal{I}_{\underline{\omega}_i}, s_j \notin \mathcal{I}_{\underline{\omega}_j}$ **do**
9:         $\sigma \leftarrow$ NEWPREDICATE
10:        $\mathcal{P} \leftarrow \mathcal{P} \,\|\, \sigma$ **if** SCOREEFF$(\sigma, \mathcal{P}, \omega, \mathcal{D})$
11:     **end while**
12: **end for**
13: **return** $\mathcal{P}$

---

---

**Algorithm 6** Scoring Functions for Invented Predicates

---

1: **Input:** New predicate $\sigma$, existing predicate set $\mathcal{P}$, skill $\omega$, and skill execution traces $\mathcal{D}$.
2: **Parameters:** Threshold $h$

3: **ScorePrecond:**
4: $\mathcal{P}' \leftarrow \mathcal{P} \cup \{\sigma\}$
5: $\mathcal{A}' \leftarrow$ LEARNOPERATORS$(\mathcal{D}, \mathcal{P}')$                       $\triangleright$ Hypothetical operators after including $\sigma$
6: total $\leftarrow 0$
7: valid $\leftarrow 0$
8: **for** $\langle s, \underline{\omega}, s' \rangle \in \mathcal{D}$ **do**
9:     **if** $\exists a' \in \mathcal{A}', \mathbf{o} \subseteq \mathcal{O}$, s.t. $s \in$ PRE$_{\underline{a}'}$, and $s \in I_\omega$ **then**
10:        valid $=$ valid $+ 1$
11:     **end if**
12:     total $=$ total $+ 1$
13: **end for**
14: **return** valid/total $> h$

15: **ScoreEff:**
16: $\mathcal{P}' \leftarrow \mathcal{P} \cup \{\sigma\}$
17: $\mathcal{A}' \leftarrow$ LEARNOPERATORS$(\mathcal{D}, \mathcal{P}')$
18: total $\leftarrow 0$
19: valid $\leftarrow 0$
20: **for** $\langle s, \underline{\omega}, s' \rangle \in \mathcal{D}$ **do**
21:     **if** $\exists a' \in \mathcal{A}', \mathbf{o} \subseteq \mathcal{O}$, s.t. $\Gamma_{\mathcal{P}'}(s') \setminus \Gamma_{\mathcal{P}'}(s) =$ EFF$_{\underline{a}'}$, and $s \in I_\omega$ **then**
22:        valid $=$ valid $+ 1$
23:     **end if**
24:     total $=$ total $+ 1$
25: **end for**
26: **return** valid/total $> h$

---

### B.3 OPERATOR LEARNING

---

**Algorithm 7** Learn Operators

---

1: **Input:** Skill execution traces $\mathcal{D}(\omega) = \{\langle s, \underline{\omega}, s' \rangle\}_\omega$, predicate set $\mathcal{P}$
2: **Output:** Operator set $\mathcal{A}$
3: eff_dict $\leftarrow$ defaultdict() $\qquad\qquad\qquad\qquad\qquad\qquad\qquad\qquad$ ▷ Store clustered effects
4: **for** $\langle s, \underline{\omega}, s' \rangle \in \mathcal{D}$ **do**
5: $\quad$ eff $\leftarrow \Gamma(s') \setminus \Gamma(s)$
6: $\quad$ eff_dict[eff] $\leftarrow$ eff_dict[eff] $\| (s, s')$
7: **end for**
8: $\mathcal{A} \leftarrow []$
9: **for** eff $\in$ eff_dict **do**
10: $\quad$ execution_list $\leftarrow$ eff_dict[eff]
11: $\quad$ precond $\leftarrow \Pi_{\langle s, \underline{\omega}, s' \rangle \in \text{execution\_list}} \Gamma(s)$
12: $\quad \mathcal{A} \leftarrow \mathcal{A} \| [\text{precond}, \text{eff}]$
13: **end for**
14: **return** $\mathcal{A}$

---

## C  PROPERTIES OF LEARNED SYMBOLIC MODELS

Relational predicates are the basic units of the abstract representation of the low-level state space. In this section, we characterize the conditions of the learned representations of a finite set of skills using relational predicates to support high-level planning, in the context of model learning. From here, chaining the skills is enabled by applying predicates from the representation of each low-level skill to others for clustering (described in Section 3.3).

Effective skill planning requires an accurate abstract model and grounding function. All forms of abstractions are typically lossy, i.e., while learning an abstract transition model, certain low-level environment details may not be captured. Conversely, the learned model must accurately retain the information needed to produce sound and complete plans. In this section, we characterize the conditions under which an abstract model facilitates exact, sound, and/or complete planning, for the purpose of constraining how we can *construct* such a model.

To begin formalizing the relationship between a skill $\omega \in \Omega$ and some abstract model $\mathcal{M} = (\mathcal{P}, \mathcal{A})$, we define two sets, $\boldsymbol{\alpha}_\omega$ and $\boldsymbol{\zeta}_\omega$, representing the states in which the model predicts that the skill may either be initiated (when $s \in \boldsymbol{\alpha}_\omega$) or terminated (when $s \in \boldsymbol{\zeta}_\omega$), respectively.

**Definition 3.** *Given a skill instance $\underline{\omega} \in \underline{\Omega}$ and abstract model $\mathcal{M} = (\mathcal{P}, \mathcal{A})$, we define $\boldsymbol{\alpha}_{\underline{\omega}}$ and $\boldsymbol{\zeta}_{\underline{\omega}}$ as follows:*

$$\boldsymbol{\alpha}_{\underline{\omega}} = \bigcup_{\underline{a} \in \underline{\mathcal{A}}_{\underline{\omega}}} \mathcal{G}(\underline{PRE_{\underline{a}}}) \tag{2}$$

$$\boldsymbol{\zeta}_{\underline{\omega}} = \bigcup_{\underline{a} \in \underline{\mathcal{A}}_{\underline{\omega}}} \mathcal{G}((\underline{PRE_{\underline{a}}} \setminus \underline{EFF_{\underline{a}}^-}) \cup \underline{EFF_{\underline{a}}^+}) \tag{3}$$

*For a non-instantiated skill $\omega \in \Omega$, we define $\boldsymbol{\alpha}_\omega = \cup_{\underline{\omega}} \boldsymbol{\alpha}_{\underline{\omega}}$ and $\boldsymbol{\zeta}_\omega = \cup_{\underline{\omega}} \boldsymbol{\zeta}_{\underline{\omega}}$.*

We define an exact abstract model as one that perfectly captures the initiation set and termination set of all skills. Although such a representation is infeasible to learn in practice, its properties provide an "ideal case" from which other definitions can weaken assumptions.

**Definition 4** (Exact Model). *Let $\mathcal{M} = (\mathcal{P}, \mathcal{A})$ be a model for environment $(\mathcal{S}, \mathcal{T}, \Omega, T)$ and the set of skills $\Omega$, and let $\alpha_\omega$ and $\zeta_\omega$ be the approximate initiation and termination sets (Def. 3). The model $\mathcal{M}$ is **an exact model** iff:*

$$\forall \omega \in \Omega, s \in \mathcal{S}: s \in \mathcal{I}_\omega \iff s \in \boldsymbol{\alpha}_\omega \text{ and } \beta_\omega(s) = 1 \iff s \in \boldsymbol{\zeta}_\omega. \tag{4}$$

An exact planning model supports accurate planning as it precisely characterizes skills' initiation and termination sets. However, this accuracy comes at the cost of practical feasibility, as any exact model achieves very little in terms of *abstraction*: it must express the full initiation and termination

sets of each skill. Therefore, in many settings, alternative model properties that approximate the exactness of the learned model, namely *soundness*, *suitability*, and *correctness*, may be preferred as objectives for model learning. We now define these properties.

A symbolic model is *sound* if it correctly predicts the effects of a plan: whenever a complete and sound planner predicts that a sequence of skills will reach some abstract state, executing the corresponding skills in the environment truly leads there. Soundness rules out spurious symbolic transitions that do not correspond to realizable outcomes. Formally, we define soundness as follows: Let $\mathcal{M} = (\mathcal{P}, \mathcal{A})$ denote a symbolic planning model, where $\mathcal{P}$ is a finite set of predicates defining an *abstract state space* $\bar{\mathcal{S}}$, and $\mathcal{A}$ is a set of *abstract actions (skills)* with preconditions and effects expressed in terms of $\mathcal{P}$. Each abstract state $\bar{s} \in \bar{\mathcal{S}}$ is obtained by a learned *grounding function* $\Gamma : \mathcal{S} \to \bar{\mathcal{S}}$ that maps low-level agent states $s \in \mathcal{S}$ to truth assignments over $\mathcal{P}$.

**Definition 5** (Soundness). *The model $\mathcal{M}$ is sound iff, for any valid plan $\pi$ produced by a complete symbolic planner over $\mathcal{M}$ and for all task instances $\mathbf{p}_i \in \mathbf{P}$,*

$$\Gamma(\mathcal{T}(\pi, s_0)) \;=\; \bar{\mathcal{T}}_{\mathcal{M}}(\pi, \Gamma(s_0)),$$

*where $s_0$ is the initial state, $\mathcal{T}(\pi, s_0)$ is the set of states reachable by executing $\pi$ from $s_0$, and $\bar{\mathcal{T}}_{\mathcal{M}}(\pi, \Gamma(s_0))$ is the abstract state predicted by $\mathcal{M}$ after executing $\pi$ from $\Gamma(s_0)$.*

A symbolic model is *complete* if it never omits real solutions: whenever the environment admits a way to solve a task, the planner can find a corresponding abstract plan in the model. Completeness rules out gaps in symbolic coverage that would make feasible problems appear unsolvable.

**Definition 6** (Completeness). *The model $\mathcal{M} = (\mathcal{P}, \mathcal{A})$ is complete if, for any task instance $\mathbf{p}_i \in \mathbf{P}$ and any sequence of low-level actions that achieves the goal from an initial state $s_0$, there exists a symbolic plan $\pi$ over $\mathcal{M}$ such that*

$$\Gamma(\mathcal{T}(\pi, s_0)) \models G_i,$$

*where $G_i$ is the goal condition of $\mathbf{p}_i$.*

A symbolic model is *suitable* if it correctly characterizes *when* a skill can be applied, meaning the symbolic preconditions predicted by the model align with the skill's real initiation conditions. A skill is applicable in an abstract state *iff* it is applicable in the corresponding grounded state.

**Definition 7** (Suitability). *The model $\mathcal{M}$ is suitable if, for any valid plan $\pi$ produced by a complete symbolic planner and for all task instances $\mathbf{p}_i \in \mathbf{P}$,*

$$\Gamma\big(\mathcal{T}(\underline{a_i}, s_0)\big) \in \bar{I}_{\underline{a_{i+1}}} \iff \mathcal{T}(\omega_i, s_0) \in I_{\omega_{i+1}}, \quad \forall a \in \mathcal{A}, \forall \omega \in \Omega$$

*where $s_0$ is an initial agent state, $\mathcal{T}(\pi, s_0)$ is the state reached by executing $\pi$ from $s_0$, and $\bar{I}_a$ is the abstract initiation set of abstract skill $a$.*

The minimum requirement for a model $\mathcal{M}$ to solve abstract planning problems is that it is *suitable* and *complete*, such that an abstract plan can always be found, and that it is always executable, if there exists a skill plan as a solution. Although methods for constructing such models exist and have been investigated in previous work, they do not provide these guarantees.

## D  PROOFS

**Lemma 2** (Predicate invention). *Let $\mathcal{D}$ be the set of transition tuples $\langle s, \omega, s' \rangle$ with $s, s' \in \mathcal{S}$ and $\omega \in \Omega$. Let $B_n$ be a buffer of $n$ i.i.d. samples from $\mathcal{D}$, each labeled with outcome and containing at least one successful and one failed transition. Let $\mathcal{M}_n = (\mathcal{P}_n, \mathcal{A}_n)$ be the model learned by SKILLWRAPPER from $B_n$, where each operator in $\mathcal{A}_n$ corresponds to some skill $\omega \in \Omega$. Then, for every $\omega \in \Omega$ appearing in $B_n$,*

$$I_\omega \triangle \boldsymbol{\alpha}_\omega = \emptyset, \tag{5}$$

$$\beta_\omega \triangle \boldsymbol{\zeta}_\omega = \emptyset, \tag{6}$$

*where $\alpha_\omega = \bigcup_{a \in \mathcal{A}_n^\omega} \mathcal{G}(\mathrm{PRE}_a)$, $\zeta_\omega = \bigcup_{a \in \mathcal{A}_n^\omega} \big(\mathcal{G}(\mathrm{PRE}_a) \setminus \mathcal{G}(\mathrm{EFF}_a^-)\big) \cup \mathcal{G}(\mathrm{EFF}_a^+)$, and $\triangle$ is the symmetric difference.*

*Proof sketch.* Suppose for contradiction that $\exists \langle s, \omega, s' \rangle \in B_n$ with $s \in I_\omega \triangle \boldsymbol{\alpha}_\omega$. Two cases arise:

**(i) False positive.** $s \notin I_\omega$ but $s \in \alpha_\omega$. By construction of $\alpha_\omega$, there must also exist $s_j \in I_\omega \cap \alpha_\omega$. Thus $1_{\alpha_\omega}(s) = 1_{\alpha_\omega}(s_j)$ while $I_\omega(s) \neq I_\omega(s_j)$, contradicting the update rule.

**(ii) False negative.** $s \in I_\omega$ but $s \notin \alpha_\omega$. Then some $s_j \notin I_\omega \cap \alpha_\omega$ must also exist, yielding the same contradiction.

Hence, no such $s$ exists, and Eq. (6) holds. The proof for Eq. (7) is identical, replacing $\alpha_\omega$ with $\zeta_\omega$ defined from operator effects. $\qquad\square$

**Theorem 3** (**Probabilistic-completeness of** SKILLWRAPPER). *Let $\mathcal{M}^*$ be a **Complete Model** for a set of skills $\Omega$, where each $\omega \in \Omega$ has initiation set $I_\omega \subseteq \mathcal{S}$ and termination set $\beta_\omega \subseteq \mathcal{S}$. Let $\mu$ be a probability distribution over $\mathcal{S} \times \mathcal{A} \times \mathcal{S}$. Consider a finite hypothesis class $\mathcal{H}$, where each $\mathcal{M} \in \mathcal{H}$ assigns a learned initiation set $\widehat{I}_\omega = \bigcup_{a \in \mathcal{A}_\omega} \mathcal{G}(\text{PRE}_a)$ and termination set $\widehat{\beta}_\omega = \bigcup_{a \in \mathcal{A}_\omega} \left( \mathcal{G}(\text{PRE}_a) \setminus \mathcal{G}(\text{EFF}_a^-) \right) \cup \mathcal{G}(\mathcal{G}_a^+)$ to each $\omega \in \Omega$.*

*For any $\mathcal{M} \in \mathcal{H}$, define*

$$d_{\text{compl}}(\mathcal{M}, \mathcal{M}^*) = \Pr_{(s, \omega, s') \sim \mu} \left[ (s \in \widehat{I}_\omega \triangle I_\omega) \vee (s' \in \widehat{\beta}_\omega \triangle \beta_\omega) \right].$$

*Let $n$ i.i.d. samples $\{(s_i, \omega_i, s_i')\}_{i=1}^n$ be drawn from $\mu$. Then, for every $\epsilon > 0$,*

$$\Pr \left[ d_{\text{compl}}(\widehat{\mathcal{M}}_n, \mathcal{M}^*) \leq \epsilon \right] \geq 1 - |\mathcal{H}| \, e^{-n\epsilon},$$

*i.e., with high probability, $\widehat{\mathcal{M}}_n$ misses fewer than an $\epsilon$-fraction of feasible transitions under $\mu$.*

*Proof sketch.* For $\mathcal{M} \in \mathcal{H}$, define the *true error*

$$\text{Err}(\mathcal{M}) = d_{\text{compl}}(\mathcal{M}, \mathcal{M}^*),$$

and the *empirical error*

$$\widehat{\text{Err}}(\mathcal{M}) = \frac{1}{n} \sum_{i=1}^n \mathbf{1} \left[ (s_i \in \widehat{I}_\omega \triangle I_\omega) \vee (s_i' \in \widehat{\beta}_\omega \triangle \beta_\omega) \right].$$

By Lemma 2, $\widehat{\text{Err}}(\widehat{\mathcal{M}}_n) = 0$. Suppose some $\mathcal{M} \in \mathcal{H}$ satisfies $\text{Err}(\mathcal{M}) \geq \epsilon$. Then each sample has probability at least $\epsilon$ of revealing an error. The chance of seeing none in $n$ i.i.d. draws is at most $e^{-n\epsilon}$ (by Hoeffding/Chernoff).

Applying a union bound over all $\mathcal{M} \in \mathcal{H}$,

$$\Pr \left[ \exists \mathcal{M} \in \mathcal{H} : \text{Err}(\mathcal{M}) \geq \epsilon \wedge \widehat{\text{Err}}(\mathcal{M}) = 0 \right] \leq |\mathcal{H}| \, e^{-n\epsilon}.$$

Since $\widehat{\mathcal{M}}_n$ has $\widehat{\text{Err}} = 0$, the event $\text{Err}(\widehat{\mathcal{M}}_n) \geq \epsilon$ is contained in this bound. Thus, with probability at least $1 - |\mathcal{H}| e^{-n\epsilon}$, $d_{\text{compl}}(\widehat{\mathcal{M}}_n, \mathcal{M}^*) \leq \epsilon$. $\qquad\square$

**Theorem 4** (**Soundness of** SKILLWRAPPER). *Let $\mathcal{T}$ be the set of transition tuples $\langle s, \omega, s' \rangle$ with $s \in \mathcal{S}$ and $\omega \in \Omega$. Let $B_n$ be an experience buffer of $n$ samples drawn from $\mathcal{T}$, each labeled with outcome. Suppose SKILLWRAPPER learns a model*

$$\mathcal{M}_n = (\mathcal{P}_n, \mathcal{A}_n),$$

*where each $a \in \mathcal{A}_n$ corresponds to some skill $\omega \in \Omega$. Then, for every operator $a \in \mathcal{A}_n$ associated with $\omega$, there exists a real transition $\langle s, \omega, s' \rangle \in B_n$ such that*

$$\underline{s} \models \text{PRE}_a \quad and \quad \underline{s}' \models \text{EFF}_a.$$

*Proof sketch.* By construction, SKILLWRAPPER derives each operator $a \in \mathcal{A}_n$ from transitions in $B_n$ through its wrapper procedure. If $a$ is associated with skill $\omega$, then its preconditions $\text{PRE}_a$ are obtained from the abstract representation of some observed $s$, and its effects $\text{EFF}_a$ from the corresponding $s'$. Hence there must exist $\langle s, \omega, s' \rangle \in B_n$ such that $\underline{s} \models \text{PRE}_a$ and $\underline{s}' \models \text{EFF}_a$.

If no such transition existed, then $a$ would be unsupported by data and would not have been generated. Thus every operator in $\mathcal{A}_n$ corresponds to at least one valid observed transition, proving soundness. $\qquad\square$

**Proof (by Contradiction).** Assume, for contradiction, that there is an operator $a \in \mathcal{A}_n$ for which *no* transition $\langle s, \omega, s' \rangle \in B_n$ supports it. That would mean:

$$(1) \ a \text{ is in the learned model,}$$

$$(2) \ a \text{ has no real sample } \langle s, \omega, s' \rangle \in B_n \text{ s.t. } \underline{s} \models \text{PRE}_a \wedge \underline{s}' \models \text{EFF}_a.$$

However, SKILLWRAPPER introduces or refines operators *only* in response to observed transitions $\langle s, \omega, s' \rangle$ that cannot be explained by any existing operator in $\mathcal{A}_n$. Therefore, if $a$ exists in the final model, it must have been created when the system encountered a transition $\langle s, \omega, s' \rangle$ with $\underline{s}$ and $\underline{s}'$ not accounted for by any previously existing operator. That transition becomes the "anchor" for $a$'s preconditions and effects.

Hence, there *must* be at least one real transition $\langle s, \omega, s' \rangle \in B_n$ matching the preconditions and effects of $a$, contradicting assumption (2). Consequently, our assumption is false, and each operator indeed has a supporting transition in $B_n$. This completes the proof. □

### D.1 ILLUSTRATION OF PREDICATE INVENTION IN LOW-LEVEL SPACE

Following the last section, we illustrate how SKILLWRAPPER invents new predicates under all circumstances with guarantees using the condition. We discuss the case of precondition here, and guarantees of effect follow the same logic.

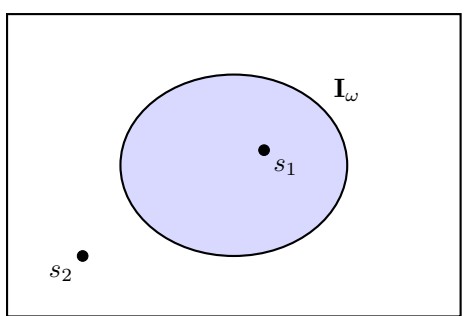

By assumption, there exist two transitions: $\langle s_1, \omega, s_1' \rangle, \langle s_2, \omega, s_2' \rangle$ such that $s_1 \in I_\omega, s_2 \notin I_\omega$ initially.

There are three possible circumstances of the resulting learning model from $\langle s_1, \omega, s_1' \rangle$ and $\langle s_2, w, s_2' \rangle$. For each of them, we discuss all possible cases of more transitions with the starting state falling into each section of the state space.

**(1).** The learned model has $I_\omega \subset \alpha$. Then, for each section that initial states of new transitions can fall in:

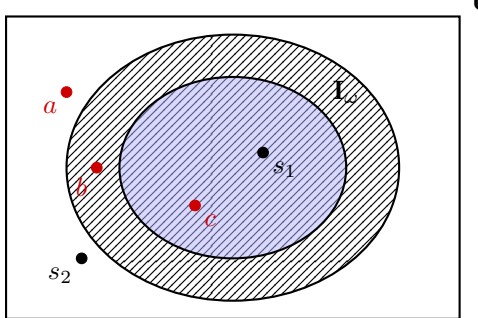

- **(a):** $a \notin \alpha, a \notin I_\omega$.
  Thus, $\nexists \ \langle s, \omega, s' \rangle$ such that $\mathbf{1}_\alpha(a) = \mathbf{1}_\alpha(s)$ while $I_\omega(a) \neq I_\omega(s)$. No additional predicate need to be invented.

- **(b):** $b \in \alpha, a \notin I_\omega$.
  Thus, $\mathbf{1}_\alpha(b) = \mathbf{1}_\alpha(s_1)$ while $I_\omega(a) \neq I_\omega(s)$. New predicate will be invented.

- **(c):** $a \in \alpha, a \in I_\omega$.
  Thus, $\nexists \ \langle s, \omega, s' \rangle$ such that $\mathbf{1}_\alpha(a) = \mathbf{1}_\alpha(s)$ while $I_\omega(a) \neq I_\omega(s)$. No additional predicate need to be invented.

**(2).** The learned model has $\alpha \subset I_\omega$. Then, for each section that initial states of new transitions can fall in:

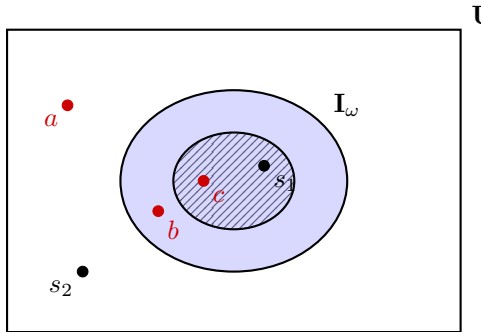

- **(a)**: $a \notin \alpha, a \notin I_\omega$.
  Thus, $\nexists \langle s, \omega, s' \rangle$ such that $\mathbf{1}_\alpha(a) = \mathbf{1}_\alpha(s)$ while $I_\omega(a) \neq I_\omega(s)$. No additional predicate need to be invented.

- **(b)**: $b \notin \alpha, a \in I_\omega$.
  Thus, $\mathbf{1}_\alpha(b) = \mathbf{1}_\alpha(s_1)$ while $I_\omega(a) \neq I_\omega(s)$. New predicate will be invented.

- **(c)**: $a \in \alpha, a \in I_\omega$.
  Thus, $\nexists \langle s, \omega, s' \rangle$ such that $\mathbf{1}_\alpha(a) = \mathbf{1}_\alpha(s)$ while $I_\omega(a) \neq I_\omega(s)$. No additional predicate need to be invented.

**(3).** The learned model has $\alpha \cap I_\omega \neq \emptyset, \alpha \nsubseteq I_\omega, I_\omega \nsubseteq \alpha$. Then, for each section that initial states of new transitions can fall in:

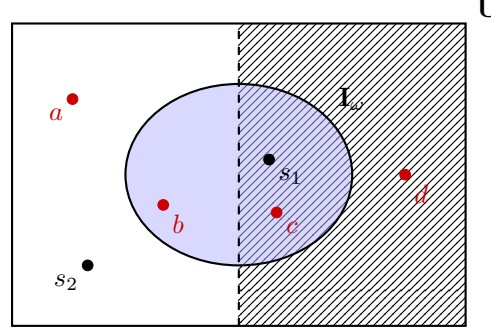

- **(a)**: $a \notin \alpha, a \notin I_\omega$.
  Thus, $\nexists \langle s, \omega, s' \rangle$ such that $\mathbf{1}_\alpha(a) = \mathbf{1}_\alpha(s)$ while $I_\omega(a) \neq I_\omega(s)$. No additional predicate need to be invented.

- **(b)**: $b \notin \alpha, a \in I_\omega$.
  Thus, $\mathbf{1}_\alpha(b) = \mathbf{1}_\alpha(s_1)$ while $I_\omega(a) \neq I_\omega(s)$. New predicate will be invented.

- **(c)**: $a \in \alpha, a \in I_\omega$.
  Thus, $\nexists \langle s, \omega, s' \rangle$ such that $\mathbf{1}_\alpha(a) = \mathbf{1}_\alpha(s)$ while $I_\omega(a) \neq I_\omega(s)$. No additional predicate need to be invented.

- **(d)**: $d \in \alpha, a \notin I_\omega$.
  Thus, $\mathbf{1}_\alpha(d) = \mathbf{1}_\alpha(s_1)$ while $I_\omega(d) \neq I_\omega(s)$. New predicate will be invented.

So far, we have discussed all possible cases where initial states of new transitions can fall in, and the predicate invention condition is proven to handle all cases.

# E    ADDITIONAL DETAILS ON THE HYPOTHESIS CLASS AND SAMPLE COMPLEXITY

Recall that our learned symbolic model has the form

$$\mathcal{M} = (\mathcal{P}, \mathcal{A}),$$

where $\mathcal{P}$ is a set of predicates and $\mathcal{A} = \bigcup_{\omega \in \Omega} \mathcal{A}_\omega$ is a set of abstract operators, with $\mathcal{A}_\omega$ the set of operators associated with skill $\omega \in \Omega$.

Each operator $a \in \mathcal{A}_\omega$ is defined by its preconditions and add/delete effects:

$$a \equiv (\omega, \Theta_a, \text{PRE}_a, \text{EFF}_a^+, \text{EFF}_a^-), \quad \text{PRE}_a, \text{EFF}_a^+, \text{EFF}_a^- \subseteq \mathcal{P}.$$

In our implementation, the number of VLM calls is finite, and the resulting models use a small number of predicates in environments with finitely many objects. For the *theoretical analysis*, we make this implicit resource bound explicit:

- Let $P_{\max}$ denote a fixed maximum number of predicates that SKILLWRAPPER is allowed to invent.

- Let $\mu_{\max}$ denote the maximum arity of any predicate $\sigma \in \mathcal{P}$.

Because SKILLWRAPPER learns one operator per lifted effect cluster, we can derive an upper bound on the number of operators per skill $\omega \in \Omega$ based on the number of possible effect sets. For any operator $a \in \mathcal{A}_\omega$ and predicate $\sigma \in \mathcal{P}$, there are three possible cases: $\sigma \in \text{EFF}_a^+$, $\sigma \in \text{EFF}_a^-$, or $\sigma \notin \text{EFF}_a^- \cup \text{EFF}_a^+$. Because the upper bound of possible instances of $p$ is $|\mathcal{O}|^{\mu_{\max}}$, we can express the maximum number of operators as

$$A_{\max} = 3^{P_{\max} \cdot |\mathcal{O}|^{\mu_{\max}}}$$

We therefore define the hypothesis class analyzed in Theorem 2 as

$$\mathcal{H} = \Big\{ (\mathcal{P}, \mathcal{A}) \,\Big|\, |\mathcal{P}| \le P_{\max}, \ |\mathcal{A}_\omega| \le A_{\max} \ \forall \omega \in \Omega \Big\}. \tag{7}$$

Predicate re-evaluation and removal during learning do *not* expand $\mathcal{H}$; they only move the learned model within this resource-bounded class by altering which predicates and operators are actively used.

### E.1 Upper Bound on $|\mathcal{H}|$

We now derive a practical upper bound on the size of $\mathcal{H}$ in equation 7.

Fix a predicate set $\mathcal{P}$ with $|\mathcal{P}| \le P_{\max}$. For each operator $\alpha$, its symbolic definition is given by three subsets of $\mathcal{P}$:

$$\textsc{Pre}_a, \ \textsc{Eff}_a^+, \ \textsc{Eff}_a^- \subseteq \mathcal{P}.$$

We consider negative precondition, so there are three possibilities for one predicate $p$: $p \in \textsc{Pre}_a$, $-p \in \textsc{Pre}_a$, and $p \notin \textsc{Pre}_a$. Since $\textsc{Eff}_a^+$ and $\textsc{Eff}_a^-$ are considered in $A_{\max}$, a single operator has at most

$$3^{P_{\max} \cdot |\mathcal{O}|^{\mu_{\max}}} \tag{8}$$

distinct configurations of preconditions.

For a fixed skill $\omega \in \Omega$, we allow at most $A_{\max}$ operators. Treating each of the $A_{\max}$ operator "slots" as independently choosing one of the $3^{P_{\max} |\mathcal{O}|^{\mu_{\max}}}$ possible configurations in equation 8, the total number of operator-sets $\mathcal{A}_\omega$ for that skill is bounded by

$$\left( 3^{P_{\max} \cdot |\mathcal{O}|^{\mu_{\max}}} \right)^{A_{\max}} = 3^{P_{\max} A_{\max} |\mathcal{O}|^{\mu_{\max}}}. \tag{9}$$

Across all skills $\omega \in \Omega$, we obtain the bound

$$|\mathcal{H}| \le \prod_{\omega \in \Omega} 3^{P_{\max} A_{\max} |\mathcal{O}|^{\mu_{\max}}} = 3^{P_{\max} A_{\max} |\Omega| |\mathcal{O}|^{\mu_{\max}}}. \tag{10}$$

### E.2 Sample Complexity for a Target $(\epsilon, \delta)$

Theorem 2 states that, for any $\epsilon > 0$,

$$\Pr\Big[ d_{\mathrm{compl}}\big(\widehat{\mathcal{M}}_n, \mathcal{M}^*\big) > \epsilon \Big] \le |\mathcal{H}| \, e^{-n\epsilon}, \tag{11}$$

where $\widehat{\mathcal{M}}_n$ is the model returned by SkillWrapper after observing $n$ i.i.d. transitions, and $d_{\mathrm{compl}}$ is the completeness distance defined in the main text (probability of a "missed feasible" event under the transition distribution).

Substituting the bound on $|\mathcal{H}|$ from equation 10 into equation 11 yields

$$\Pr\Big[ d_{\mathrm{compl}}\big(\widehat{\mathcal{M}}_n, \mathcal{M}^*\big) > \epsilon \Big] \le 3^{P_{\max} A_{\max} |\Omega| |\mathcal{O}|^{\mu_{\max}}} \, e^{-n\epsilon}. \tag{12}$$

To guarantee that this probability is at most $\delta \in (0, 1)$, it suffices that

$$3^{P_{\max} A_{\max} |\Omega| |\mathcal{O}|^{\mu_{\max}}} \, e^{-n\epsilon} \le \delta,$$

which is equivalent to

$$n \ge \frac{1}{\epsilon} \Big( P_{\max} A_{\max} |\Omega| |\mathcal{O}|^{\mu_{\max}} \ln 3 + \ln \tfrac{1}{\delta} \Big). \tag{13}$$

Thus, the number of transitions required to ensure

$$d_{\mathrm{compl}}\big(\widehat{\mathcal{M}}_n, \mathcal{M}^*\big) \le \epsilon \quad \text{with probability at least } 1 - \delta$$

is

$$n = \mathcal{O}\left(\frac{P_{\max}A_{\max}|\Omega||\mathcal{O}|^{\mu_{\max}} + \log(1/\delta)}{\epsilon}\right). \tag{14}$$

In our experiments, the arity and realized numbers of predicates are small. Setting $P_{\max}$ to match the practical budget yields numerical values in equation 13 for the regimes we study. Our theoretical result therefore formalizes how increasing the capacity of the symbolic model (via larger $P_{\max}$) trades off against the number of transitions needed to achieve a desired completeness level.

## F  Learned Operators, Case Studies, and Example Tasks

### F.1  Learned Operators

```
# Example Learned Predicate and Operator of Burger domain

# Predicate
 name: cut_into_pieces
 types:
 - cuttable
 semantic: "the cuttable appears as at least two non-touching visible pieces (has multiple
 disconnected regions), indicating it is cut rather than a single intact piece."

# Operator
(:action Cut_7
 :parameters (?cuttable_p0 - cuttable
           ?pickupable_p1 - pickupable
           ?pickupable_p2 - pickupable
           ?pickupable_p6 - pickupable
           ?robot_p5 - robot
           ?station_p4 - station)
 :precondition (and
   (not (= ?pickupable_p1 ?pickupable_p2))
   (not (= ?pickupable_p1 ?pickupable_p6))
   (not (= ?pickupable_p2 ?pickupable_p6))
   (gripper_empty ?robot_p5)
   (on_cutting_board ?cuttable_p0)
   (on_station ?cuttable_p0 ?station_p4)
   (top_most ?cuttable_p0)
   (not (cut_into_pieces ?cuttable_p0))
   (not (holding ?robot_p5 ?cuttable_p0))
   (not (holding ?robot_p5 ?pickupable_p1))
   (not (holding ?robot_p5 ?pickupable_p2))
   (not (stacked_on ?pickupable_p6 ?cuttable_p0)))
 :effect (and
   (cut_into_pieces ?cuttable_p0)))
```

```
# Example Learned Predicate and Operator of Franka domain

# Predicate
 name: plate_is_dirty
 types:
 - plate
 semantic: the specified plate's upper surface contains visible food traces whose appearance
 differs from the plate's base surfac (visible residue to be wiped).

# Operator
(:action Stack_5
 :parameters (?pickupable_p0 - pickupable
           ?pickupable_p1 - pickupable
           ?pickupable_p3 - pickupable
           ?plate_p4 - plate
           ?robot_p2 - robot)
 :precondition (and
   (not (= ?pickupable_p0 ?pickupable_p1))
   (not (= ?pickupable_p0 ?pickupable_p3))
   (not (= ?pickupable_p1 ?pickupable_p3))
   (holding ?robot_p2 ?pickupable_p1)
   (plate_top_unoccupied ?plate_p4)
   (not (gripper_empty ?robot_p2))
   (not (holding ?robot_p2 ?pickupable_p0))
   (not (holding ?robot_p2 ?pickupable_p3))
   (not (plate_is_dirty ?plate_p4))
   (not (stacked_on ?pickupable_p1 ?plate_p4)))
 :effect (and
```

```
      (gripper_empty ?robot_p2)
      (stacked_on ?pickupable_p1 ?plate_p4)
      (not (holding ?robot_p2 ?pickupable_p1))
      (not (plate_top_unoccupied ?plate_p4))))
```

```
# Example Learned Predicate and Operator of Bi-manual Kuka domain

# Predicate
 name: Coated
 types:
 - utensil
 semantic: a visible layer or clump of material adheres to the utensil's working end (e.g.,
 the blade shows a smear that was absent before).

# Operator
(:action Scoop_156
 :parameters (?openable_p0 - openable
            ?robot_p1 - robot
            ?utensil_p2 - utensil)
 :precondition (and
    (HeldByRobot ?robot_p1 ?openable_p0)
    (InLeftGripper ?robot_p1 ?openable_p0)
    (InRightGripper ?robot_p1 ?utensil_p2)
    (LidOff ?openable_p0)
    (not (Closed ?openable_p0))
    (not (Coated ?utensil_p2))
    (not (InContainer ?utensil_p2))
    (not (LeftGripperEmpty ?robot_p1))
    (not (OpenableOnTable ?openable_p0))
    (not (RightGripperEmpty ?robot_p1))
    (not (UtensilOnTable ?utensil_p2)))
 :effect (and
    (Coated ?utensil_p2)))
```

Additionally, we present here a predicate and an operator written by the PDDL expert:

```
# Predicate and Operator Written by PDDL Expert

# Predicate
 name: is_on_station
 types:
 - pickupable
 - station
 semantic: "A `pickupable` object is on top of a `station`."

# Operator
(:action Cut
 :parameters (?robot - robot
            ?cuttable - cuttable
            ?board - cuttingboard)
 :precondition (and
    (hand_empty)
    (obj_free ?cuttable)
    (is_on_station ?cuttable ?board)
    (not (is_cut ?cuttable)))
 :effect (and
    (is_cut ?cuttable)))
```

### F.2 RESULT ANALYSIS OF ROBOTOUILLE

We here present case studies of the failure modes of SKILLWRAPPER and the baselines. One common failure mode is misclassifications during inference time, where we abstract low-level states specified in raw images into symbolic states using the predicates in the learned model. Since our contributions are more on the theory and algorithm design, we did not explore various potential techniques, such as ViperGPT (Surís et al., 2023), that can possibly mitigate this issue.

**System Predicates** (Sys Preds.)  are designed in a way such that every possible state can be specified with a compact set of eight predicates, as shown in the experiments. However, this is proven insufficient to describe the model dynamics. For example, the predicate clear_above or its semantic equivalent is necessary for describing the precondition of all operators learned from the *Pick* skill, such that the model can express "Picking an item with another item on top will fail." Since predicate invention is disabled in Sys Preds., it often returns invalid plans that attempt to pick objects

that are underneath a stack. As a result, the lack of soundness induces a low solved rate for hard and impossible problems of System Predicates.

**Random exploration** is heavily limited by inefficient data gathering. However, if we take a closer look at the operators it learns, they are of good quality, which is the benefit of sharing the same predicate invention algorithm of SKILLWRAPPER. In fact, one frequent failure mode under this randomness is that the baseline may never execute the skills with complex preconditions successfully. For example, in the Robotouille setting, the *Cook* action requires the item to be on the stove and the agent's hand to be empty. Since the same predicate invention algorithm with SKILLWRAPPER is being used, it can only learn operators for the skills that have been successfully executed in the observed transitions. As a result, it only learns operators for skills that are usually executable, such as *Pick*, and it thus can only solve the simplest pick and stack tasks. Additionally, random exploration also achieves 100% on *Impossible*, yet because the learned model is not plannable.

**No Heuristic** shares a similar failure mode as System Predicates—achieves good performance in *Easy* problems but degrades significantly in *Hard*. With further investigation, we found that the baseline generally invents fewer predicates than SKILLWRAPPER, which results in occasionally missing critical ones (two out of the five total runs). This observation explains the large variances in *Hard* and *Impossible* problems, and also indicates that *Easy* problems could be solved even with an incomplete predicate set, which aligns with our findings in System Predicate. In turn, it supports the usefulness of the two engineered heuristics for skill sequence proposal. We believe improving the exploration strategy is a promising direction for future work. Another point we want to note is that either the solved rate or the planning budget only evaluates the learned operators from the planning outcome, while better metrics are needed for evaluating the exploration efficiency.

**SKILLWRAPPER**'s failure mode is similar to the case of Random exploration. Comparing the poorly performing PDDL operators learned by SKILLWRAPPER to Expert Operators, the model only contains one extra predicate, on_cutting_board(item), which divides the previous cluster that shares the same effect into even smaller clusters. Then, the transition data in these smaller clusters cannot support the model learning algorithm to effectively eliminate spurious preconditions. This case study points out that the balance between predicate invention and data gathering is a critical factor in the learning process: if too many predicates are invented without adequate transition data, the resulting operators could possibly contain spurious preconditions, such that it cannot generalize at all. In SKILLWRAPPER, the balance is controlled by empirically tuning the threshold of the scoring function and the length of each skill sequence proposed. Another sub-optimality is that our algorithm does not filter invented predicates that are semantic synonyms or antonyms to existing predicates, which increases the classification burden of the foundation model. Though these redundant predicates are usually well handled by the foundation model, a smarter prompting system could be designed to mitigate this issue and improve computation efficiency.

### F.3 EXAMPLE TASKS

Here, we provide additional examples of the tasks, using planning problem from the evaluation data. Specifically, Robotouille tasks use images generated by the simulator, Franka tasks use images taken by a fixed camera in front of the robot, and Bimanual Kuka tasks use images taken by its egocentric camera.

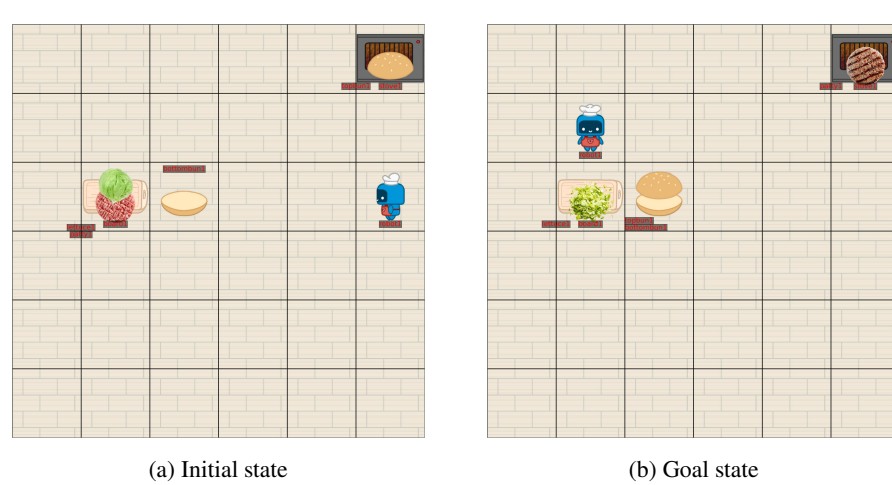

(a) Initial state          (b) Goal state

Figure 7: Example task in Robotouille.

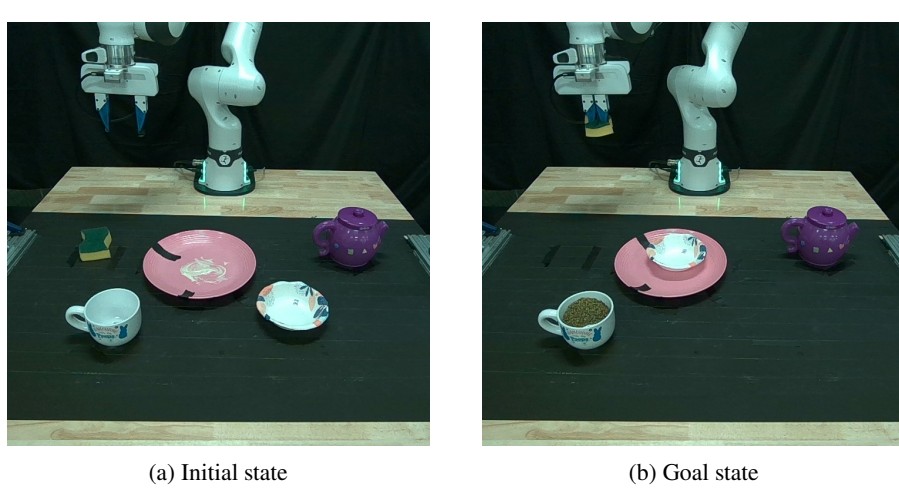

(a) Initial state          (b) Goal state

Figure 8: Example task in Franka.

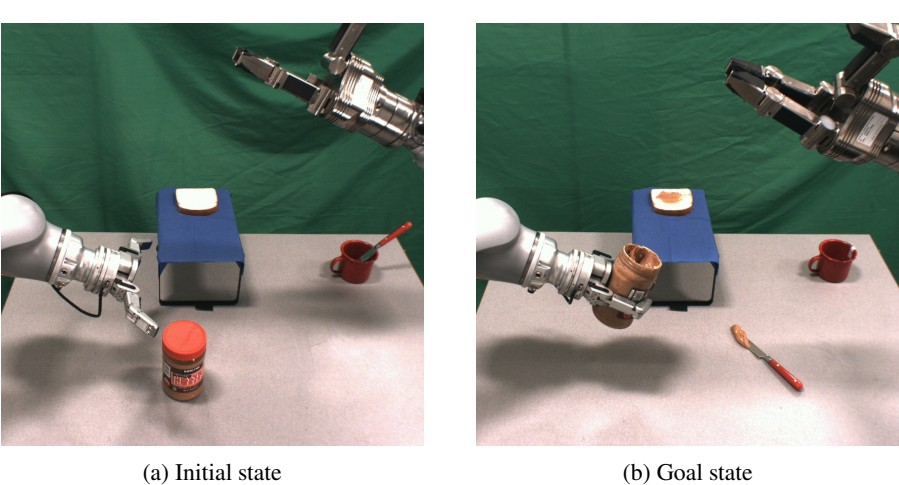

(a) Initial state          (b) Goal state

Figure 9: Example task in Bimanual Kuka.

## G    VLM Reliability Study

### G.1    Classification Accuracy

We here analyze the classification accuracy of the vision-language model (VLM) used in both robotic experiments. Since the predicates are generated on the fly by the VLM, we do not have the ground truth values for them, and thus we must verify if the truth values match the images manually.

**Per-predicate Classification Evaluation.**    Since predicates are originally lifted and can be grounded with different combinations of objects, we first define a classification over a low-level state of a grounded predicate as correct if (1) all parameters appear in the scene (if the predicate is not nullary) and (2) the truth value of the predicate match the low-level state specified by the image input. Then, we define a classification of a lifted predicate over a low-level state as correct if all of its grounded instances are classified correctly over that state.

**Results and Analysis.**    Over all predicates, the classification accuracy is $86.7\%$ for the Franka experiment, and $98.5\%$ for the bimanual Kuka experiment. Compared to the planning performance reported for both experiments, the classification accuracy is generally much higher. One reason for this mismatch is that, due to the rigidity of symbolic planning, even flipping the truth value of a single predicate can lead to a planning failure. To support this claim, we found specific poorly performing predicates that hinder the planning task the most, and we provide more quantitative results in the next paragraph.

**Per-predicate Accuracy.**    The learned symbolic model of the Franka experiment contains 6 predicates, which have 11 possible grounded instances. The learned symbolic model of the bimanual Kuka experiment contains 12 predicates, which have 13 possible grounded instances. We evaluate per-predicate accuracy for both in Table 3 and Table 4. From the results of the Franka experiment, we identify the two predicates, `gripper_empty` and `holding`, that caused all planning failures, and they fail almost simultaneously due to their semantic correlation. With further investigation, we found that the misclassifications were induced by a single object, `Sponge`, which is possibly due to the color of the object and the background being too similar. In the bimanual Kuka experiment, it is `coated` (if the knife has peanut butter on it) that caused most of the planning failure, likely caused by the lighting conditions. These observations suggest the accuracy of VLM is a limiting factor, and resolving them poses a promising path to improving the performance of SkillWrapper.

Table 3: Per-predicate accuracy of Franka.

|  | gripper_empty | holding | mug_full | plate_top_unoccupied | stacked_on | plate_is_dirty |
|---|---|---|---|---|---|---|
| Accuracy (%) | 60.0 | 60.0 | 100.0 | 100.0 | 100.0 | 100.0 |

Table 4: Per-predicate accuracy of Bimanual Kuka.

|  | InLeftGripper | InRightGripper | RightGripperEmpty | LeftGripperEmpty | LidOff | InContainer |
|---|---|---|---|---|---|---|
| Accuracy (%) | 100.0 | 100.0 | 100.0 | 100.0 | 100.0 | 98.3 |
|  | OpenableOnTable | Closed | Coated | SpreadOn | HeldByRobot | UtensilOnTable |
| Accuracy (%) | 96.7 | 100.0 | 88.3 | 98.3 | 100.0 | 100.0 |

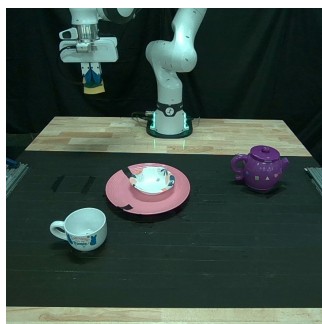
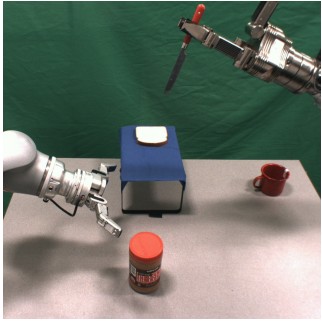

(a) ✗`holding(Sponge) = F`        (b) ✗`coated(Knife) = T`

### G.2 REAL-WORLD ROBUSTNESS

To evaluate the real-world robustness of the VLM, we additionally conduct experiments to investigate factors such as viewpoints, lighting conditions, or domain shifts. For each of them, we collect a held-out set of images by varying these factors. We report per-predicate accuracy, and all numbers are averaged across three individual runs.

**Viewpoints.** We collect visual observation data from two viewpoints and sample five configurations for each viewpoint. From the results of Franka experiments, we observe that the classification accuracies of certain predicates are higher from the viewpoint closer to the corresponding objects: at viewpoint #1, all predicates can be perfectly classified, while predicates involving gripper or mug, such as `gripper_empty`, `holding` and `mug_full`, are significantly lower from viewpoint #2, which is farther from the objects. In bimanual Kuka experiments, the result is mostly stable across different viewpoints and generally better than in the Franka environment, which is possibly due to fewer background distractions. Though the accuracy varies across viewpoints, its performance remains reliable as long as the full observability assumption still holds.

Table 5: Per-predicate accuracy of Franka at Viewpoint #1.

|  | gripper_empty | holding | mug_full | plate_top_unoccupied | stacked_on | plate_is_dirty |
|---|---|---|---|---|---|---|
| Accuracy (%) | 100.0 | 100.0 | 100.0 | 100.0 | 100.0 | 100.0 |

Table 6: Per-predicate accuracy of Franka at Viewpoint #2.

|  | gripper_empty | holding | mug_full | plate_top_unoccupied | stacked_on | plate_is_dirty |
|---|---|---|---|---|---|---|
| Accuracy (%) | 80.0 | 80.0 | 90.0 | 100.0 | 100.0 | 100.0 |

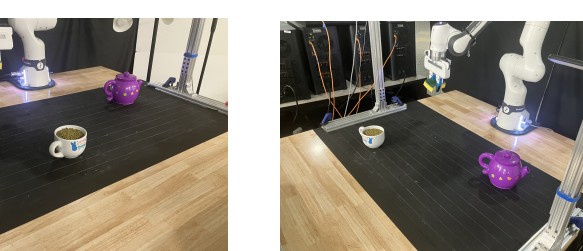

(a) ✓ `holding(Sponge) = T`    (b) ✗ `holding(Sponge) = F`

Table 7: Per-predicate accuracy of Bimanual Kuka at Viewpoint #1.

|  | InLeftGripper | InRightGripper | RightGripperEmpty | LeftGripperEmpty | LidOff | InContainer |
|---|---|---|---|---|---|---|
| Accuracy (%) | 100.0 | 100.0 | 100.0 | 100.0 | 100.0 | 100.0 |
|  | OpenableOnTable | Closed | Coated | SpreadOn | HeldByRobot | UtensilOnTable |
| Accuracy (%) | 100.0 | 100.0 | 100.0 | 100.0 | 100.0 | 100.0 |

Table 8: Per-predicate accuracy of Bimanual Kuka at Viewpoint #1.

|  | InLeftGripper | InRightGripper | RightGripperEmpty | LeftGripperEmpty | LidOff | InContainer |
|---|---|---|---|---|---|---|
| Accuracy (%) | 100.0 | 100.0 | 100.0 | 100.0 | 100.0 | 100.0 |
|  | OpenableOnTable | Closed | Coated | SpreadOn | HeldByRobot | UtensilOnTable |
| Accuracy (%) | 100.0 | 100.0 | 100.0 | 100.0 | 100.0 | 93.3 |

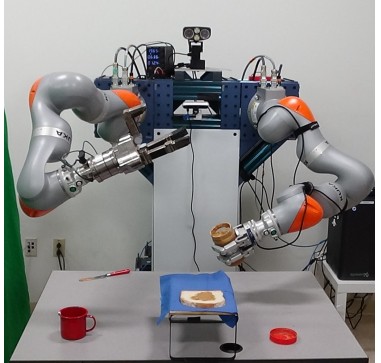 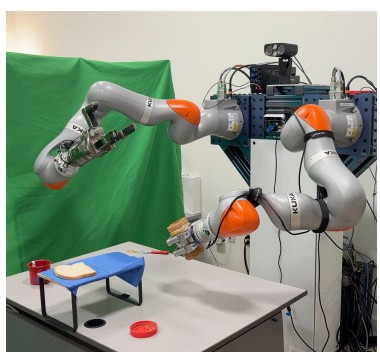

(a) ✓ UtensilOnTable(Knife) = **T**     (b) ✗ UtensilOnTable(Knife) = **F**

**Lighting conditions.** We collect visual observation data under two lighting conditions and sample five configurations for each one. We find that the VLM is generally robust to different lighting conditions, except for several extremely hard ones, such as under lighting condition #2 in Bimanual Kuka, where the objects are heavily shadowed.

Table 9: Per-predicate accuracy of Franka under Lighting Condition #1.

|  | gripper_empty | holding | mug_full | plate_top_unoccupied | stacked_on | plate_is_dirty |
|---|---|---|---|---|---|---|
| Accuracy (%) | 90.0 | 90.0 | 100.0 | 96.7 | 100.0 | 100.0 |

Table 10: Per-predicate accuracy of Franka under Lighting Condition #2.

|  | gripper_empty | holding | mug_full | plate_top_unoccupied | stacked_on | plate_is_dirty |
|---|---|---|---|---|---|---|
| Accuracy (%) | 90.0 | 90.0 | 100.0 | 98.3 | 100.0 | 100.0 |

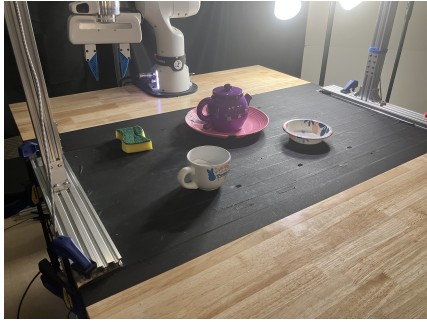 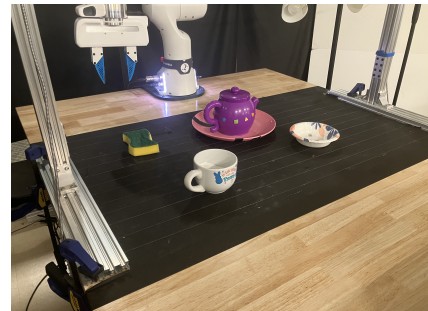

(a) ✓ stacked_on(Teapot, Plate) = **T**     (b) ✓ stacked_on(Teapot, Plate) = **T**

Table 11: Per-predicate accuracy of Bimanual Kuka under Lighting Condition #1.

|  | InLeftGripper | InRightGripper | RightGripperEmpty | LeftGripperEmpty | LidOff | InContainer |
|---|---|---|---|---|---|---|
| Accuracy (%) | 100.0 | 100.0 | 100.0 | 100.0 | 100.0 | 100.0 |
|  | OpenableOnTable | Closed | Coated | SpreadOn | HeldByRobot | UtensilOnTable |
| Accuracy (%) | 100.0 | 100.0 | 100.0 | 100.0 | 100.0 | 100.0 |

Table 12: Per-predicate accuracy of Bimanual Kuka under Lighting Condition #2.

|  | InLeftGripper | InRightGripper | RightGripperEmpty | LeftGripperEmpty | LidOff | InContainer |
|---|---|---|---|---|---|---|
| Accuracy (%) | 100.0 | 80.0 | 80.0 | 100.0 | 100.0 | 73.3 |
|  | OpenableOnTable | Closed | Coated | SpreadOn | HeldByRobot | UtensilOnTable |
| Accuracy (%) | 100.0 | 100.0 | 100.0 | 86.7 | 100.0 | 100.0 |

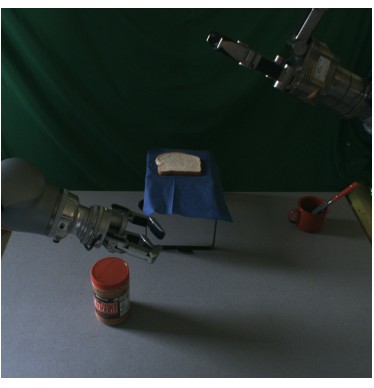 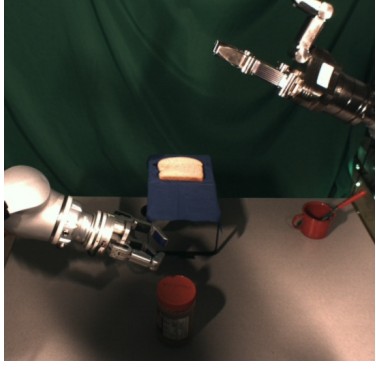

(a) ✓ `RightGripperEmpty`() = **T**  (b) ✗ `RightGripperEmpty`() = **F**

**Domain shift.** SKILLWRAPPER relies entirely on semantics to prompt the VLM, abstracting raw states into symbolic states without using visual features. The only requirement for generalizing to novel objects is that the VLM can correctly identify the object referents in the image based on their type information provided in the language prompt. To evaluate this generalization capability, we collected visual observation data (five images per environment) under domain shift by swapping objects with new instances and sampling two configurations. From this observation, we found that the only failure mode introduced by domain shifts occurs when the VLM cannot recognize an object because its visual appearance does not align with the semantics. For example, a plate that is too small might be misclassified as a saucer, leading to incorrect symbolic states.

Table 13: Per-predicate accuracy of Franka under Domain Shift.

|  | gripper_empty | holding | mug_full | plate_top_unoccupied | stacked_on | plate_is_dirty |
|---|---|---|---|---|---|---|
| Accuracy (%) | 90.0 | 90.0 | 100.0 | 100.0 | 73.3 | 100.0 |

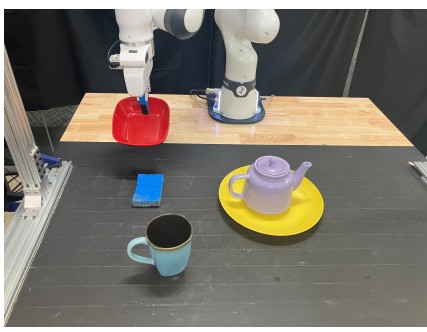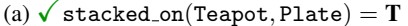 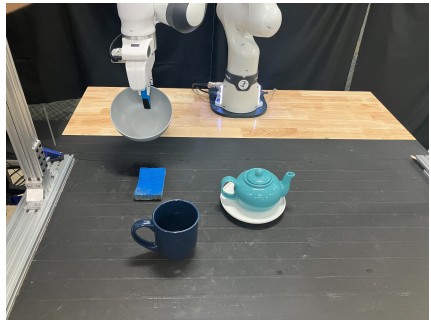

(a) ✓ `stacked_on`(Teapot, Plate) = **T**  (b) ✗ `stacked_on`(Teapot, Plate) = **F**

Table 14: Per-predicate accuracy of Bimanual Kuka under Domain Shift.

| | InLeftGripper | InRightGripper | RightGripperEmpty | LeftGripperEmpty | LidOff | InContainer |
|---|---|---|---|---|---|---|
| Accuracy (%) | 100.0 | 100.0 | 100.0 | 100.0 | 100.0 | 100.0 |
| | OpenableOnTable | Closed | Coated | SpreadOn | HeldByRobot | UtensilOnTable |
| Accuracy (%) | 100.0 | 100.0 | 100.0 | 100.0 | 100.0 | 100.0 |

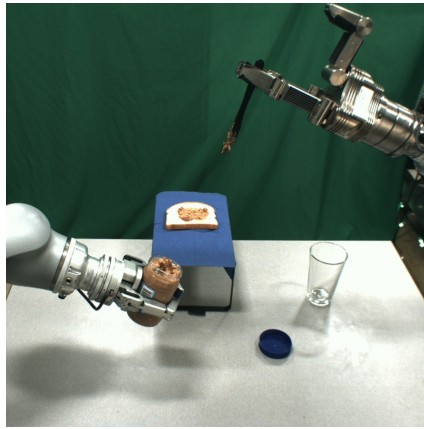 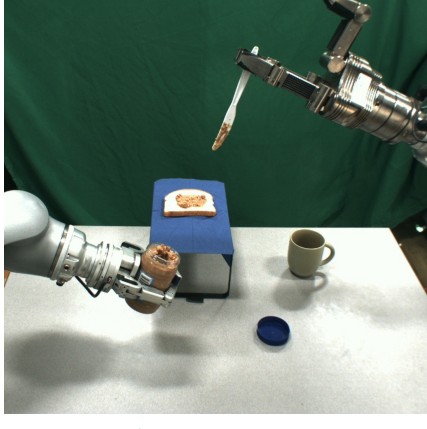

(a) ✓ coated(Knife) = **T**    (b) ✓ coated(Knife) = **T**

### G.3 OTHER VLMS

We further examine the possibility of using open-source VLMs as alternatives for SKILLWRAPPER. We choose Qwen3-VL-235B (Bai et al., 2025) for comparison. To evaluate its capability, we conduct two sets of preliminary experiments: predicate classification and predicate invention.

**Predicate Classification.** We collected a subset of images (five from Franka and ten from Bimanual Kuka) and evaluated the truth values of each predicate with the two models. From the result, we observed that the two models have different failure patterns, and a prominent one is that Qwen3 can reliably detect if the gripper is holding an object, except for occasional classification errors on the objects being held. In general, we found two models perform on par with each other, and thus we believe they can be used interchangeably for predicate classification.

Table 15: Per-predicate accuracy of Franka.

| | gripper_empty | holding | mug_full | plate_top_unoccupied | stacked_on | plate_is_dirty |
|---|---|---|---|---|---|---|
| GPT-5 Acc. (%) | 60.0 | 60.0 | 100.0 | 100.0 | 100.0 | 100.0 |
| Qwen3 Acc. (%) | 100.0 | 80.0 | 100.0 | 100.0 | 100.0 | 80.0 |

Table 16: Per-predicate accuracy of Bimanual Kuka.

| | InLeftGripper | InRightGripper | RightGripperEmpty | LeftGripperEmpty | LidOff | InContainer |
|---|---|---|---|---|---|---|
| GPT-5 Acc. (%) | 100.0 | 100.0 | 100.0 | 100.0 | 100.0 | 100.0 |
| Qwen3 Acc. (%) | 100.0 | 100.0 | 100.0 | 100.0 | 100.0 | 100.0 |
| | OpenableOnTable | Closed | Coated | SpreadOn | HeldByRobot | UtensilOnTable |
| GPT-5 Acc. (%) | 100.0 | 100.0 | 100.0 | 100.0 | 100.0 | 100.0 |
| Qwen3 Acc. (%) | 100.0 | 100.0 | 90.0 | 80.0 | 100.0 | 100.0 |

**Predicate Invention.** We qualitatively compare the performance of both models on inventing predicates by reasoning over contrastive pairs of transitions. For each environment, we curated two contrastive pairs, and each model is prompted by the same input to invent one new predicate. A

predicate is considered correct if it is a semantic synonym or antonym of the target predicate. From the result, we can conclude that GPT-5 is much more reliable in reasoning over the transitions for predicate invention, and thus Qwen3 cannot be used as an alternative for this specific task. (We omitted ?robot from all predicates' arguments for simplicity.)

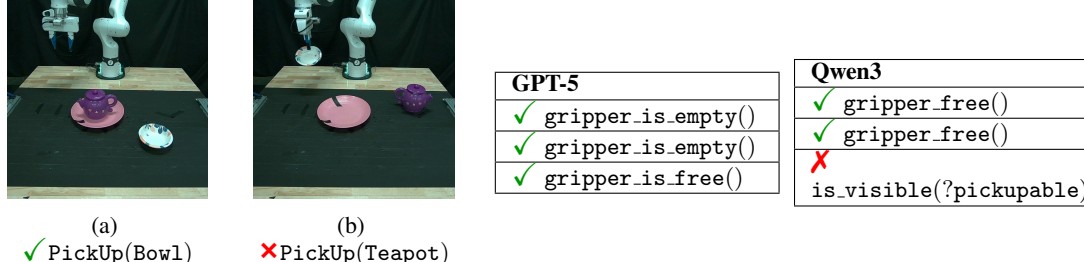

| GPT-5 | Qwen3 |
|---|---|
| ✓ gripper_is_empty() | ✓ gripper_free() |
| ✓ gripper_is_empty() | ✓ gripper_free() |
| ✓ gripper_is_free() | ✗ is_visible(?pickupable) |

(a) ✓PickUp(Bowl)  (b) ✗PickUp(Teapot)

Figure 17: **Predicate Invention Case #1 in Franka.** *Target predicate*: GripperEmpty( *Existing predicates*: ∅

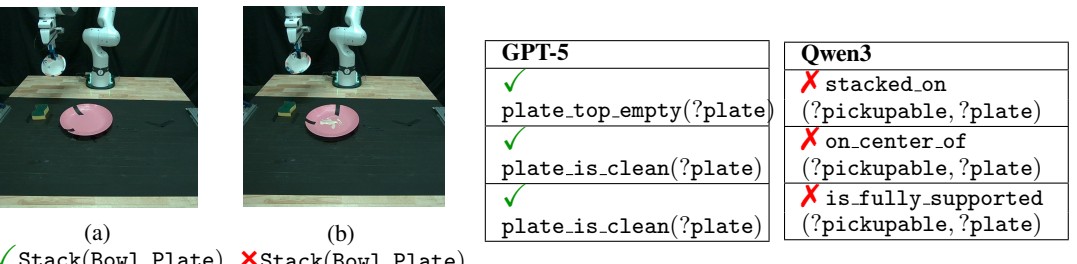

| GPT-5 | Qwen3 |
|---|---|
| ✓ plate_top_empty(?plate) | ✗ stacked_on (?pickupable, ?plate) |
| ✓ plate_is_clean(?plate) | ✗ on_center_of (?pickupable, ?plate) |
| ✓ plate_is_clean(?plate) | ✗ is_fully_supported (?pickupable, ?plate) |

(a) ✓Stack(Bowl,Plate)  (b) ✗Stack(Bowl,Plate)

Figure 18: **Predicate Invention Case #2 in Franka.** *Target predicate*: PlateIsDirty(?plate) *Existing predicates*: GripperEmpty(), Holding(?pickupable)

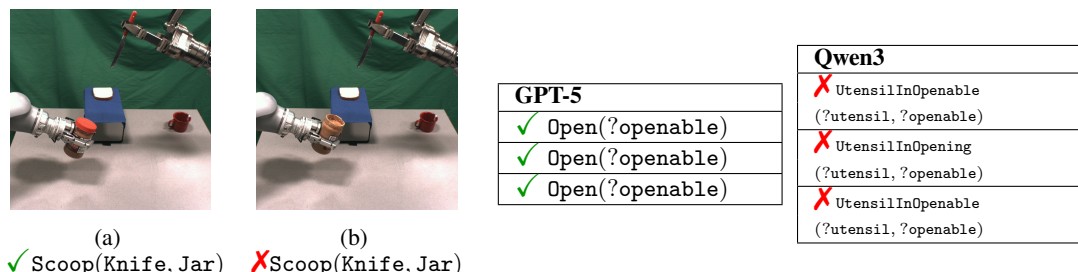

| GPT-5 | Qwen3 |
|---|---|
| ✓ Open(?openable) | ✗ UtensilInOpenable (?utensil, ?openable) |
| ✓ Open(?openable) | ✗ UtensilInOpening (?utensil, ?openable) |
| ✓ Open(?openable) | ✗ UtensilInOpenable (?utensil, ?openable) |

(a) ✓Scoop(Knife,Jar)  (b) ✗Scoop(Knife,Jar)

Figure 19: **Predicate Invention Case #1 in Bi-Kuka.** *Target predicate*: LidOff(?openable) *Existing predicates*:InLeftGripper(?openable), InRightGripper(?utensil)

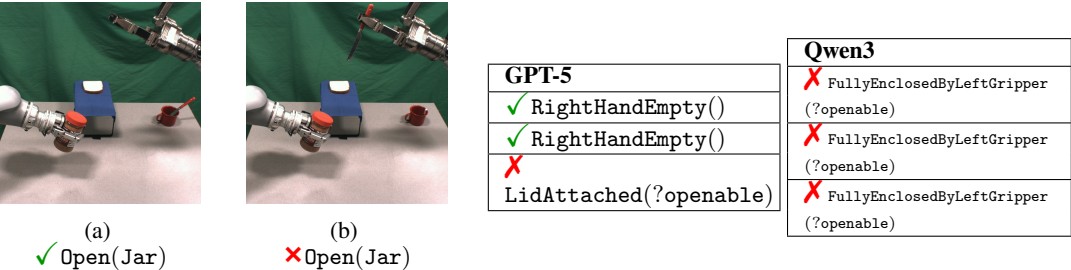

| GPT-5 | Qwen3 |
|---|---|
| ✓ RightHandEmpty() | ✗ FullyEnclosedByLeftGripper (?openable) |
| ✓ RightHandEmpty() | ✗ FullyEnclosedByLeftGripper (?openable) |
| ✗ LidAttached(?openable) | ✗ FullyEnclosedByLeftGripper (?openable) |

(a) ✓Open(Jar)  (b) ✗Open(Jar)

Figure 20: **Predicate Invention Case #2 in Bi-Kuka.** *Target predicate*: RightGripperEmpty() *Existing predicates*:InLeftGripper(?openable), LidOff(?openable)

# H IMPLEMENTATION DETAILS

## H.1 PLANNER AND PLANNING TIME

We use K* planner (Katz & Lee, 2023) to generate top $K$ optimal plans, where $K$ in practice is the maximum planning budget. We use an i9-13900F CPU for running all the planning tasks. On average, each planning problem takes $0.0599$ seconds. Specifically, in Robotouille experiments, easy problems take $0.0549$ seconds, hard problems takes $0.0583$ seconds, and impossible problems take $0.0565$ seconds per problem; in Franka experiments, in-domain problems take $0.0529$ seconds, generalization problems take $0.5175$ seconds, and impossible tasks take $0.0516$ seconds per problem; in Bimanual Kuka experiments, all problems take $0.0553$ seconds on average.

## H.2 API CALL

For running the experiments, we made roughly 9300 calls to GPT-5, which cost \$96.59 in total.

## H.3 HYPERPARAMETERS

We here report and summarize all hyperparameters of SKILLWRAPPER used for the experiment to provide better reproducibility. For all experiments, we set the batch size of skill sequence proposal to be $5$ and interaction budget per iteration to be $15$, and we run SKILLWRAPPER for $5$ iterations. For Robotouille experiments, we set the threshold $h$ to be $0.6$. For Franka and bimanual Kuka experiments, we set the threshold $h$ to be $0.5$.

## H.4 ROBOT EXPERIMENTS

**Single-Arm Manipulation.** We employ a Franka Emika Research 3 robotic arm equipped with a UMI gripper (Chi et al., 2024). The workspace is observed by a single Intel RealSense D455 exocentric RGB-D camera, oriented to capture both the tabletop scene and the robot. The RGB data from this camera are used for learning symbolic models, while the depth information supports object pose estimation. Object poses are estimated using FoundationPose (Wen et al., 2024), which leverages high-fidelity 3D scanned models of the target objects. System-level communication and coordination are implemented in ROS 2 (Humble), which interfaces with motion planning, perception, and control modules. This setup supports five parameterized skills: *Pick*, *Place*, *Stack*, *Pour*, and *Wipe*. The first four skills (*Pick*, *Place*, *Stack*, and *Pour*) are executed through motion planning with the MoveIt framework, conditioned on both the end-effector and object poses. The *Wipe* skill is implemented by replaying a teleoperated trajectory.

**Bimanual Manipulation.** We use a robot with two horizontally mounted KUKA LBR iiwa 7 R800 manipulators, one with a BarrettHand BH8-282 gripper, and the other with a Schunk Dextrous Hand 2.0 gripper. The robot collects RGB data used for learning symbolic models with a MultiSense S7 camera mounted on a Pan-Tilt unit, while using an Intel RealSense D455 camera for RGB-D data used in pose estimation (Wen et al., 2024) of the objects in the scene. We use ROS 1 and KUKA FRI to communicate with the robot and utilize the built-in joint impedance control with position target as the low-level controller. At the high level, we create collision models of all objects in the scene and use a task and motion planner to generate motion plans for each skill. The *Pick* skills (compatible with knife and peanut butter jar) are implemented using motion planning. The *OpenJar*, *Scoop*, and *Spread* skills are implemented using a combination of motion planning and pre-defined trajectory playback.

## H.5 LANGUAGE MODEL PROMPTS

In this section, we provide the prompts used for the core components of SKILLWRAPPER (specifically skill sequence proposal, predicate invention, and predicate evaluation) as well as the ViLA (Hu et al., 2023) baseline. For predicate evaluation (Appendix H.8), we empirically observed that it is more accurate when evaluation is done in batches, where the truth values of multiple predicates are evaluated at once rather than one at a time. In addition, when asking for a fixed and structured output, the accuracy is significantly lower than a free-form output. Therefore, we adopt a two-stage evaluation process: in the first stage, the foundation model generates a response in any format, and in the second stage, it provides a summary of the output from the previous step.

## H.6 Skill Sequence Proposal

---

**System Prompt**

<AGENT_DESCRIPTION> is attempting to learn the preconditions and effects for a finite set of skills by executing exploratory skill sequences and exploring the environment.

---

**Skill Sequence Proposal Prompt**

Propose a set of skill sequences for a robot to execute. The robot is attempting to learn the preconditions and effects for a finite set of operators. The robot can navigate the environment freely but only has one gripper. The robot has access to the following skills with their associated arguments:

[SKILL_PROMPT]

The list of objects the robot has previously encountered in the environment are:

[OBJECT_IN_SCENE]
[ENV_DESCRIPTION]

The pairs of consecutive skills (skill1, skill2) that have been least explored are: [[LEAST_EXPLORED_SKILLS]]. Certain skills have similar names and arguments, but different preconditions and effects. Using the list of objects and the skill preconditions / effects learned, generate 5 skill sequences and their sequence of skills such that:
(1) the skill sequences should violate their preconditions occasionally.
(2) at least 1 unexplored skill pair is used in each skill sequence.
(3) all skill sequences have at least 15 skills in sequence.
(4) there are no same skills with same arguments consecutively in the sequence.

Output only the sequence of skills to execute, ensuring to follow the naming/syntax/arguments for skills provided. Output 1 skill every new line, following the format below:

Skill Sequence 1:
GoTo(CounterTop)
PickUp(Apple, CounterTop)

Skill Sequence 2:

---

## H.7 PREDICATE INVENTION

---

**Predicate Invention Prompt**

[AGENT_DESCRIPTION]

The robot has been programmed with the skill [LIFTED_SKILL]
two times. In the first execution, the grounded skill
[GROUNDED_SKILL_1] [SUCCESS_1], and in the second execution,
[GROUNDED_SKILL_2] [SUCCESS_2]. The difference in outcomes suggests
that the existing predicate set is insufficient to fully capture
the preconditions for successful execution of this skill.

Your task is to propose a single new high-level predicate and its
semantic meaning based on the visual comparison of the two input
images taken before each execution.

Predicates should meet these criteria:
- The predicate must be grounded in visual state only (e.g.,
"gripper is open," "object is above table," "arm is holding
object").
- Describe object state or spatial relations relevant to task
success (e.g., gripper open/closed, object on left/right of
gripper, object touching/supporting another object, etc.)
- Do not infer properties like affordances (is_graspable), alignment
with grippers, or success likelihood that are vaguely defined and
cannot be clearly determined visually.
- Avoid using concept like grasping zone or robot's reachability to
define the predicate since they are not defined by common sense.
- Use at most 2 parameters (e.g., predicate(x), predicate(x,
y), predicate()), where robot arm must be included for any
robot-environment relation.
- Avoid predicates that assume internal properties like
is_graspable, is_properly_aligned, or any accessibility/reachability
reasoning that cannot be determined visually.
- The semantic meaning should be a grounded and objective
description of the predicate in terms of the physical scene (e.g.,
"the object is fully enclosed by the robot's gripper"), not about
execution success or skill dynamics.
- The parameters of the predicate must be subset of the parameters
of the skill.

Format your output as follows:
`predicate_name(parameters)`: semantic_meaning.

for example:
`CloseTo(arm, location)`: the robot arm is close to the location.

Current predicates: [PRED_LIST]

Previously proposed but rejected predicates: [TRIED_PRED]

Avoid duplicates or near-duplicates of existing predicates and
rejected predicates. Reason over using a paragraph and generate
the predicate and the semantic meaning in the given format in a
separate line.

One new predicate candidate for improving the representation of the
precondition for [LIFTED_SKILL] ( Don't use any parameter other than
[PARAMETERS]):

---

## H.8 PREDICATE EVALUATION

---

**Predicate Evaluation Prompt: Step 1**

```
Given the current observation of the simulated kitchen domain,
the object types, and the list of predicates, what are the true
grounded predicates?

[ENVIRONMENTAL_DESCRIPTION]

Objects:
[OBJECTS]

Predicates:
[PREDICATES]
```

---

**Predicate Evaluation Prompt: Step 2**

```
Summarize what the true grounded predicates are from this response,
and list them in the format of predicate_name(arg1, arg2, ...)  in
separate lines with no any formatting.  If the response contains
typos of object names or redundant indices, you should correct
them.  Correct object names are:  [OBJECT_NAMES]. If the response
include redundant predicates that are not in this list, you should
filter them.  Correct predicates are:  [PRED_NAMES]. The response
is:

"""
[RESPONSE]
""
```

---

## H.9 VILA

---

**ViLA Prompt**

```
You are [AGENT_DESCRIPTION]. As a robot, you are able to execute the
following skills:
[SKILLS]

Here are the objects and their types that are compatible with your
skills:
[OBJECTS]

You are given two images:  The first one captures your current
observation, and the second one specifies your goal.  Given both
images, your job is to generate a plan starting from the *current
state* to the goal state.  You should first reason about the goal
of the task and how the skills can be chained to solve it in the
first paragraph.  After the reasoning, return the plan from the
current state in a new paragraph by listing skills in separate
lines with no additional explanation, header, or numbering.
Use "Done" in the skill list to indicate the task is complete,
and report if the task is impossible to solve by simply returning
"Impossible".
```

---

## I    RELATED WORKS

**Skill Abstraction.**    There has been a long track of works focusing on building hierarchies that abstract away high-dimensional details with low-dimensional abstractions for planning (Konidaris & Barto, 2009; Konidaris et al., 2018; Shah et al., 2024), and those applied to robotics are usually connected to task and motion planning (TAMP) (Shah et al., 2020; Garrett et al., 2021). These approaches, however, are incapable of handling high-dimensional sensory-motor signals (such as images) as input. Research on action model learning (Xi et al., 2024; Juba et al., 2021) learn symbolic action models for input skills. However, unlike our method, these approaches require symbols to be provided as input. Similar to our system's integration of self-play and focus on uncovering skill conditions, Verma et al. (2022) focus on assessing capabilities of black-box agents for grid world-like tasks while assuming that the agent is an oracle. A tangential research effort on chaining various skills in novel environments involves training extra models (Yokoyama et al., 2024) and STRIPS task planner with action primitives (Gu et al., 2022; Szot et al., 2021).

**Predicate Learning for Robotic Tasks.**    Predicates provide a convenient way to abstract away low-level details of the environment and build efficient and compact representations. Prior to foundation models, previous attempts to build classifiers for predicates from raw image inputs originated from the neuro-symbolic domain (Johnson et al., 2017; Mao et al., 2019), and their initial application for robotics took a similar supervised learning approach with labeled demonstrations (Migimatsu & Bohg, 2022) or generated tasks (Lamanna et al., 2023). After the emergence of foundation models, recent works guide skill learning with predicates generated by LLM or together with human interaction. Li et al. (2024) invents symbolic skills for reward functions used for RL training but cannot generalize to skills learned through latent objectives, which is more commonly seen in imitation learning. Li & Silver (2023) and Han et al. (2024) leverage human experts to provide feedback to the LLM to help it improve the learned predicates and skills.

**Task Generation for Robotics.**    The approach of automatically proposing tasks has been studied for active learning and curriculum learning in grid worlds and games (Wang et al., 2019; Jiang et al., 2021) to robotic domains (Fang et al., 2021; 2022). Lamanna et al. (2023) generates tasks in PDDL as training sets to learn classifiers for object properties in predicates format, while they assume the action operators are given. With the commonsense reasoning ability of foundation models, recent works have applied the idea of automatic task proposing and self-playing for exploration (Nasiriany et al., 2024; Ren et al., 2024), data collection (Wang et al., 2024c; Yang et al., 2024; Ahn et al., 2024), boosting skills learning (Ha et al., 2023; Wang et al., 2024a), and scene understanding (Jiang et al., 2025). These works indicate a promising direction for generating robotic data and scaling up. Following the idea, we equipped our system with a task-proposing module for generating skill sequences specific to skills and predicates, which serves the idea of both data collection and exploration.

**Embodied Reasoning with Foundation Models.**    There has been a track of work on leveraging large language models (LLMs) for embodied decision-making (Huang et al., 2022; Raman et al., 2024) and reasoning (Huang et al., 2023), while vision-language models (VLMs) are often considered to have limited embodied reasoning ability due to their pre-training corpora that focus primarily on language generation (Valmeekam et al., 2023). Common ways of addressing this issue include fine-tuning on datasets from a specific domains (Hong et al., 2023; Mu et al., 2024; Chen et al., 2024) or knowledge distillation (Sumers et al., 2023; Yang et al., 2024). Meanwhile, many works manage to leverage preexisting models without further training from direct visual observation (Fang et al., 2024; Nasiriany et al., 2024) to complete robotic tasks (Jiang et al., 2025). In these works, the embodied reasoning ability of the foundation models serves as the central part of the systems. However, most benchmarking works evaluate the embodied reasoning ability of the models in a question-answering fashion (Sermanet et al., 2024; Majumdar et al., 2024; Cheng et al., 2024; Chen et al., 2025), where it remains unclear whether they are capable of solving robotic tasks.

## J    USE OF LARGE LANGUAGE MODELS

Our work incorporates language models as part of SKILLWRAPPER, particularly for the three important components of our system discussed in Section 3. We utilize OpenAI's GPT-5 (OpenAI, 2025) as

our foundation model of choice. We acknowledge that all the content of the manuscript has been generated by the authors. However, we have used LLMs for basic editing, polishing, and grammar checking.

