# OpenReview forum: "SkillWrapper: Generative Predicate Invention for Skill Abstraction"
_ICLR.cc/2026/Conference — Submitted to ICLR 2026_

### Official Review · Reviewer_eiUT · 2025-10-26

**Soundness:** 3
**Presentation:** 3
**Contribution:** 3
**Rating:** 6
**Confidence:** 3

**Summary:**

The paper tackles the problem of learning plannable, high-level models of black-box skills from raw RGB observations by inventing predicates and composing them into PDDL operators that can be used with off-the-shelf planners. The core idea is a formal framework for generative predicate invention with explicit target properties (soundness, completeness, “suitability”), and an algorithm, SKILLWRAPPER, that alternates between (i) actively collecting data by proposing skill sequences, (ii) inventing predicates when current abstractions cannot explain observed success/failure or effects, and (iii) learning operators by clustering transitions on lifted effects and intersecting preconditions. The system uses a foundation model both to generate predicate candidates and to classify their truth values from images; only RGB images are assumed. Theoretical results show operators are supported by data (soundness) and that the learned model is probabilistically complete under finite-hypothesis assumptions. Empirically, the method is evaluated in Robotouille (a simulated kitchen domain) and on two real setups (Franka Panda; bimanual Kuka). In simulation, SKILLWRAPPER achieves 73.3% solved on “Easy” with PB=2.9, 38.3% on “Hard” with PB=6.1, and 100% correct detection of impossible tasks, outperforming ViLa and random exploration and competitive with expert operators on some splits. On real robots, it generalizes predicates learned in restricted settings to a larger test setting (e.g., 60.0% solved in generalization split; PB=4.0). Iterative runs show performance improves as more predicates are invented and data collected.

**Strengths:**

Originality.

 • Provides a formal target for predicate invention in the context of skill abstraction, not merely state abstraction, and uses it to drive algorithmic design (two concrete invention triggers based on executability and effect inconsistencies).
 • Uses a foundation model as a relational classifier (truth assignment for grounded predicates) and not just a planner, enabling learning from RGB images without pre-defined state factors.

Quality (theory).

 • Clear statements of soundness (operators backed by observed transitions) and probabilistic completeness w.r.t. a distribution over transitions, with proofs/sketches and finite-hypothesis assumptions.

Quality (empirics).

 • Evaluations span simulation and two real robot platforms; the setup probes both generalization across environments (Franka) and iterative improvement under irreversible actions (bimanual Kuka). The Robotouille benchmark comparison includes expert, system predicates, ViLa, and random exploration. Metrics include solved rate and a planning budget proxy (PB) tied to completeness.

Clarity.

 • The paper is organized around a tight loop: data → predicates → operators, with pseudocode, invention conditions, and scoring functions, plus prompt details in the appendix. Examples of learned predicates/operators (with natural-language semantics) help assess interpretability.

Significance.

 • Demonstrates that language/VLM priors can yield plannable, interpretable abstractions from raw images, and that these abstractions can scale to longer-horizon problems than open-loop LLM planning—an important step for integrating FM-driven perception with symbolic planning in robotics.

**Weaknesses:**

Theoretical scope and assumptions.
 • The soundness guarantee is empirical (“supported by at least one observed transition”) and does not formalize robustness to classification noise from the VLM. In practice, VLM truth assignments will be imperfect; the theory currently does not propagate uncertainty through operator learning or planning, nor does it bound error from misclassifications.
 • The probabilistic completeness bound relies on a finite hypothesis class H and i.i.d. sampling; it is unclear how H is instantiated in practice when predicate proposals are open-ended (FM-generated) and when pruning/reevaluation can expand or contract the model class over time. The bound risks being vacuous without a concrete characterization of |H| or sample complexity as a function of invented predicates.
 • The method assumes deterministic skills and that skills affect only the bound objects. Many real skills are stochastic and produce side effects; the invention conditions and operator learning rules may need adaptation to handle such cases.

Algorithmic design choices.
 • Operator learning computes preconditions via intersection of initial abstract states within a cluster; this is conservative and can produce spurious preconditions when data are sparse—acknowledged by the authors—but the paper offers limited guidance on when predicate re-evaluation suffices versus when additional data are required. A quantitative analysis of false-positive preconditions over iterations would strengthen the claim.
 • Predicate selection hinges on thresholds in score functions (Algorithm 6). The paper does not study sensitivity to the threshold h, nor how choices trade off compactness vs. coverage (e.g., learned operator sparsity, PB, solved rate).
 • The active data collection relies on LLM-proposed sequences and heuristic scores (coverage/chainability). While well-motivated, there is no ablation isolating the gain from these heuristics vs. random or simpler curricula.

Empirical evaluation.
 • Baselines: “System Predicates” lacks invention and has privileged state access; by construction it will underperform on tasks requiring new predicates, making it a weak comparator. Missing are stronger learned-predicate/action-model baselines (e.g., neurosymbolic predicate learners or prior predicate-invention techniques) to more precisely attribute gains to the proposed invention logic.
 • Scale and variance: Many results are averaged over three runs; the real-robot evaluation uses small problem sets, limiting statistical confidence. Reporting confidence intervals and significance tests would help.
 • VLM-as-classifier reliability: Since abstract states are inferred directly from a VLM on RGB images, experiments should report truth-assignment accuracy against labeled ground truth (even on a subset) and robustness to viewpoint changes/occlusion; currently, the paper assumes full observability from images.

**Questions:**

Questions
1. Hypothesis class & bounds. How do you instantiate the finite hypothesis class H used in Theorem 2 when predicates are FM-generated and can be reevaluated/removed? Can you provide a practical upper bound on |H| (e.g., as a function of max invented predicates and arity) and a sample-complexity estimate in transitions to achieve a target \epsilon?
2. Noise-aware guarantees. Do you foresee modifying the framework to incorporate noisy predicate truth values (e.g., via probabilistic predicates or confidence-weighted effects), and can the soundness/completeness results be extended to this setting? Empirically, what is the observed misclassification rate of the VLM over your predicates?
3. Ablations. Please provide ablations for (a) coverage/chainability in skill-sequence proposal, (b) predicate re-evaluation (on/off), and (c) threshold h in Algorithm 6. How do these affect solved %, PB, number of predicates, and operator sparsity?
4. Stochastic or side-effectful skills. How does the effect-inconsistency trigger behave if a skill succeeds but produces variable effects (or mid-execution failure) due to stochasticity? Do you anticipate inventing context predicates vs. effect predicates, and how are these disambiguated empirically?
5. Portability across embodiments. In the real-robot experiments, to what extent were the same predicates reused across Panda and Kuka (vs. reinvented)? Could you report a transfer study where operators learned on one platform are applied to the other with minimal additional data?
6. VLM classifier robustness. Have you measured sensitivity to camera viewpoint/lighting/occlusion or to minor domain shift (e.g., new mugs/utensils not seen during learning)? Even a small held-out labeled set would be informative.
7. Metrics. Beyond solved% and PB, could you report plan optimality gaps, planning time, interaction budget, and predicate/operator counts over iterations (with variance), to better illuminate data efficiency and model compactness?

---

> ### Author Response · Authors · 2025-11-18
>
> We would like to thank the reviewer for the insightful comments and constructive questions.
>
> __Weaknesses:__
>
> 1. __Theoretical scope and assumptions.__
>     - (a) __Soundness guarantee is empirical.__ We defined the property of a sound model in a theoretical way. However, because of the data-driven nature of SkillWrapper’s learning process, its learned abstractions can only be empirically validated based on observed data. Requiring that each learned operator be supported by at least one valid transition guarantees that every operator is executable in at least one observed low-level state. Without additional assumptions about the problem setting, we cannot make statements about the soundness of learned planning abstractions with respect to unseen states.
>
>     - (b) __Formal guarantees do not model VLM classification noise.__ Please see our answer to Q2 and General Response 3.
>
>     - (c) __Hypothesis class for prob.__ completeness bounds. Please see our answer to Q1.
>
>     - (d) __SkillWrapper assumes deterministic skills.__ Please see General Response 1.
>
>     - (e) __Skills only affect the bound objects.__ We would like to clarify that this assumption is made for the object-centric skills over the low-level state space, but it does not apply to the invented predicates and learned operators. For example, if a bottle is initially on a table and `PickUp(Bottle)` is executed, we assume that the skill only affects the low-level state of the bottle, but not the table. However, in the symbolic state, SkillWrapper will consider truth value changes of nullary predicates such as `hand_free()`, unary predicates such as `is_holding(?object)`, and binary predicates that contain any skill parameters, such as `is_on(?object, ?table)`. Therefore, SkillWrapper’s operator learning algorithm can capture conditional effects, provided that the relevant predicates are nullary or involve at least one skill parameter in the relevant grounded abstract transitions. Generally, we believe that it is reasonable to assume that well-defined object-centric skills would affect only the low-level state of their object arguments.
>
> 2. __Algorithmic design choices.__
>      - (a) __Limited guidance on when predicate re-evaluation suffices versus when additional data is required.__ We would first like to clarify that predicate re-evaluation only happens after additional data is collected, so they are not alternatives to each other. We believe that the underlying question is, “How do we know when the operators are ‘good enough’ to stop collecting data?” We believe this question is not specific to our framework but broadly applies to any learn-from-data setting and is inherently difficult. Our current framework can prove the incorrectness of a model with respect to observed data, but cannot definitively prove model correctness. Achieving that would require additional assumptions about the state distributions during training and testing, which we view as an open question and important direction for future work.
>
>     - (b) __Other weaknesses in algorithmic design choices.__ We believe that the other two weaknesses with respect to algorithmic design essentially fall under empirical evaluation and will be resolved by running the suggested ablation studies.
>
>     - (c) __How choices on threshold h trade off compactness vs. coverage (e.g., learned operator sparsity, PB, solved rate).__ We would like to kindly ask the reviewer to clarify their intended definitions of metrics for compactness, coverage, and operator sparsity, so we can accurately address this question. Intuitively, we understand “compactness” as corresponding to the number of predicates in an operator, and “coverage” as a measure of the classification accuracy of an operator’s preconditions. With this understanding, we do not believe there is a trade-off between these two values, as our scoring function (Alg. 6) considers “predicate usefulness” and model correctness, while ignoring operator compactness.
>     - (d) __No ablation study on active data collection versus random or simpler curricula.__ We would like to kindly point out that the random exploration baseline is already included in the Robotouille experiments. Please check Sec. 4.2 and Table 1 for details. (Failure case study of random exploration can be found in Appendix F.2.)

---

> ### Author Response · Authors · 2025-11-18
>
> 3. __Empirical evaluation.__
>     - (a) __System predicates baseline.__ The “System Predicates” baseline uses the built-in predicate set of the Robotouille simulator, which is designed to define any possible simulated state unambiguously. We believe it is not unsurprising that such a well-designed predicate set is insufficient to construct a complete symbolic model, as it performs as well as expert operators in simple problems. We think the baseline appears weak due to these missing details, and we have updated its description in the manuscript.
>     - (b) __Other Baselines.__ We believe that VisualPredicator is the closest work to our paper, but it is still incomparable to SkillWrapper due to its multiple strong assumptions (Detailed comparisons can be found in our response to reviewer 2cbF). We did not find prior work described as “neurosymbolic predicate learners.” We are open to incorporating baseline comparisons with comparable prior work if the reviewer could kindly provide the relevant references.
>
>      - (c) __Scale and Variance.__ We have updated the manuscript to include standard deviations. Also, we are running the experiments with two additional seeds, and we will report the results as soon as we have them.
>
>     - (d) __VLM reliability.__ We are running the proposed experiments. Please see general responses 2 and 3.
>
>     - (e) __Full observability.__  Please see general response 1.
>
> __Questions:__
>
> 1. __Hypothesis class & bounds.__ We thank the reviewer for their interest in strengthening our theoretical contribution and guidance on the derivation. We have provided additional details on the upper bound of $|H|$ and a sample complexity estimate in Appendix E.
>
> 2. __Noise-aware guarantees.__ We see two natural extensions of our framework to a noise-aware setting in future work. First, one can replace deterministic predicates with probabilistic ones, where each grounded predicate is associated with a confidence score, and operators are learned and applied using these confidences (e.g., via thresholds or probabilistic preconditions/effects). Second, the soundness and completeness notions in our paper can be relaxed to probabilistic variants (e.g., $(\alpha,\delta)$-soundness and $(\epsilon,\delta)$-completeness) that bound the probability that noisy predicate evaluations lead to incorrect operator application or missed feasible skills. Empirically, we have analyzed the VLM classification accuracy and updated it in the appendix. Please see General Response 3 for a summary and Appendix G for more details.
>
> 3. __Ablations.__ We are actively running the suggested ablation studies. However,  we would like to highlight that experiments ablating over "predicate re-evaluation" and "sensitivity to $h$" would not be fruitful because of the following reasons:
>    - **Predicate Re-evaluation**: Firstly, by re-evaluating, we mean re-scoring previously invented predicates with the new data. We consider this as one of the key parts of the algorithm that is necessary to guarantee soundness of the learned abstractions with respect to newly collected data, and therefore, feel it would be unfair to ablate over it (We will clarify this in the manuscript).
>     - **Sensitivity to $h$**: We would also like to note that we view $h$ as a standard hyperparameter. Although exploring a full sensitivity analysis would be interesting, extensive tuning is not central to the scientific questions of the paper. Here, we selected one of three initial values of $h$ (0.5, 0.6, and 0.7) per environment without heavy tuning; notably, these default values already produced strong results across our experiments. We observed that different values for $h$ do not change the finally invented predicates and operators, but mainly affect the number of iterations needed. Values for $h$ and other hyperparameters can be found in Appendix H.3.

---

> ### Author Response · Authors · 2025-11-18
>
> 4. - (a) __Skill side effects.__ We would like to kindly point out that skills with side-effects (or conditional effects) are handled by the system. SkillWrapper handles conditional effects by associative model learning: transitions with different lifted effects, even for the same skill instance, are clustered into distinct partitions, each corresponding to a learned operator. The preconditions of each operator are derived by taking the intersection of the unified initial states of all transitions in its cluster.
>
>     - (b) __Stochastic skills.__ Please see General Response 1 for more details.
>
>     - (c) __Context predicates and effect predicates.__ We believe that by “context predicates,” the reviewer means predicates that are used as preconditions of operators learned from effect clusters containing conditional effects, and by “effect predicates,” they mean the effects of those operators. Both types of predicates will be handled by the predicate invention conditions, as answered in Q4(a). Though we can infer the usefulness of predicates (as either a precondition or effect) using the scoring function, our algorithm does not necessarily have to disambiguate the two for predicate invention or operator learning.
>
> 5. - (a) __Portability across embodiments.__ The learned predicates are essentially classifiers learned over a set or distribution of in-domain states. Though SkillWrapper learns predicates with human-interpretable semantics, which may enable transfer under certain conditions, the system was not explicitly designed for transferability, and we thus do not make claims of their transferability. For example, the predicate “Nearby(object)” may affect skill feasibility differently on a robot with a short arm and a robot with a long arm.
>
>     - (b) __Transferring Study.__ We agree that it is an interesting future work to investigate the conditions under which predicates or operators can be successfully transferred to novel domains or embodiments, but we believe this is beyond the scope of this paper.
>
> 6. __VLM classifier robustness.__ We are currently running the suggested experiments on viewpoint/lighting variations and minor domain shift. However, we feel that **occlusions** violate our core assumption of fully-observable state spaces. Therefore, it would not be fair to the approach to consider such settings. Please see General Response 2 for more details of the experiments.
>
> 7. __Metrics.__ We have updated the requested metrics in the paper: interaction budget is added to the second paragraph of Sec. 4.2, first paragraph of Sec. 4.3, and Appendix H.3, planning time is updated in Appendix H.1, and others are updated accordingly to the result tables and figure 6. However,  we feel the experiments on optimality are not in the scope of the current work as we do not claim to be optimal.

---

> > ### Author Response · Authors · 2025-12-04
> >
> > We have now completed all proposed experiments in response to the reviewer: VLM classification test on viewpoint/lighting and minor domain shift (__Q6__), Table 1 with 5 individual runs (__W3.(c)__), and ablation for skill sequence proposal heuristics (__Q3__). The results are incorporated and updated in the manuscript. Please see our latest general response for the details. We believe that we have fully answered and addressed all questions raised by the reviewer.

---

### Official Review · Reviewer_FNJ2 · 2025-10-29

**Soundness:** 3
**Presentation:** 1
**Contribution:** 2
**Rating:** 2
**Confidence:** 3

**Summary:**

The authors propose a formal theory to characterize the necessary conditions for generative predicate invention, and present the SKILLWRAPPER framework, which leverages the capability of LLMs to perform operator learning and skill abstraction. Both simulation and real-world robot experiments are conducted to validate the framework.

**Strengths:**

1.The paper provides a theoretical proof of completeness for skill learning. Although I did not go through the detailed derivation, I believe this is likely one of the main innovations of the paper.

2.The paper includes real-robot experiments, which convincingly demonstrate that the SKILLWRAPPER framework can be effectively deployed in real-world settings.

**Weaknesses:**

1. In Table 1, the experimental results of SKILLWRAPPER do not consistently outperform expert-designed predicates and operators. This raises concerns about whether the proposed framework offers a real advantage over manually defined predicates and operators.
2. The experimental section lacks sufficient task descriptions. It is difficult to understand how task difficulty is defined or differentiated. If such details exist, please indicate where they are presented.
3. The authors only evaluate on the Robotouille simulated task. Why not conduct experiments on more well-known benchmarks such as IsaacGym (https://github.com/isaac-sim/IsaacGymEnvs), MetaWorld(https://github.com/Farama-Foundation/Metaworld)? Please explain the rationale behind this choice.
4. The paper’s main contributions are not clearly highlighted. Both skill abstraction and predicate generation using LLMs are not entirely new techniques. The paper lacks comparisons with these baselines, and the overall presentation fails to make the core novelty of the work clear.

**Questions:**

1. Why does ViLA not include the Planning Budget metric? Please provide a rough explanation.
2. How are the conclusions derived from the formal theory used to guide the design of the SKILLWRAPPER framework?
3. Has any ablation study been conducted—for example, how would removing active data collection affect the skill abstraction process? What would happen if expert-designed operators and SKILLWRAPPER-learned skills were combined?
4. Other questions are mentioned in the Weaknesses section.

---

> ### Author Response · Authors · 2025-11-18
>
> We would like to thank the reviewer for the comments and questions.
>
> Firstly and importantly, we would like to kindly point out that we have summarized our contributions in the last paragraph of Sec. 1, which we believe accurately describes our core novelty.
>
> Secondly, though mentioned as a strength, we would like to clarify that we do not study skill learning in this paper, and our proof includes both completeness and soundness for skill abstraction instead of skill learning.
>
> **Weaknesses:**
>
> 1. __Not consistently outperforming expert operators.__ We would like to clarify that *expert-designed predicates and operators* function as oracle knowledge and therefore serve as an upper bound for what a learning-based system can reasonably achieve. The central advantage of our approach is that it reduces dependence on such domain experts and enables robots to **invent predicates and operators tailored to their own embodiment, capabilities, and task distributions**. This is practically necessary: there are simply not enough “robot experts” available to manually engineer abstractions for every domain in which robots must operate. Prior work has emphasized this need and explored similar directions [1,2,3,4,5]. Importantly, our method not only matches expert-provided abstractions but **exceeds** them in several settings, underscoring that even highly skilled roboticists struggle to hand-design effective abstractions for complex domains. Designing such abstractions requires deep expertise not only in robotics but also in the specific task domain, and individuals with this combination are exceedingly rare. We hope this addresses the reviewer’s concern.
>
> 2. __Description of task difficulty.__ We approximate the task difficulty in Robotouille in terms of the length of what would have been an optimal plan for a task. The details could be found in the first paragraph of Sec. 4.2. We have also added example tasks of each domain in Appendix F.3.
>
> 3. __No experiments on IsaacGym or MetaWorld.__ We thank the reviewer for their suggestion for alternative simulation environments. However, we believe that these are not well-suited for evaluating our system. Specifically, Meta-World and IsaacGym are platforms for RL policy learning tasks, which are not relevant to our work. SkillWrapper assumes agents equipped with pre-defined skills for solving complex, long-horizon tasks. To better elaborate, a task in Meta-World may solely involve training a policy for a single high-level action of opening a cupboard, but our real-world robot tasks are long-horizon and involve sequences of such high-level actions. We believe that the two sets of experiments on real robots are better indicators of the real-world applicability of SkillWrapper, since the complexity of solving tasks in the real world is strictly higher than any simulated environment, such as lighting conditions, background distractors, etc. For Robotouille, we think of it as a common and handy platform that allows future methods to compete with, compared to environments and robot settings that are locally available to us. We have also provided all training and evaluation environments to facilitate this process.
>
> 4. - (a) __Main contributions & Baselines.__ We believe that our contributions are highlighted and precisely described in the last paragraph of the introduction section. For baselines, we would like to kindly ask for appropriate baselines that the reviewer feels we may have missed, because no specific baselines are mentioned by the reviewer.
>
>     - (b) __Overall presentation.__ To consolidate a constructive discussion, we would like to kindly ask the reviewer to elaborate on where the presentation has fallen short. We welcome each and every suggestion that would improve the quality of the paper, and we are more than happy to incorporate them.

---

> ### Author Response · Authors · 2025-11-18
>
> __Questions:__
>
> 1. __ViLa doesn’t have planning budgets.__ The planning budget metric is designed for evaluating only the methods that use symbolic planners: with a planner, we generate $k$ plans for one problem, where $k$ is equal to the total planning budget, and we then try iteratively executing each of the plans starting with the top 1. If a valid plan is executed, i.e., executing the plan leads to the goal state, we terminate the iteration and report the number of plans that have been tried as its planning budget. Though we provide ViLa max steps and let it take three attempts, ViLa as a VLM-based method does not have a planning budget, because it does not use a symbolic planner for planning.
>
> 2. __How the formal theory guides the design of SkillWrapper.__  We believe this question could be better addressed by reading Sec. 3 and Appendices B, C, and D, while we here sketch the connections between our theory and system: we first characterize the conditions for a learned model to be sound and complete. Then, we design the predicate invention conditions and the scoring function, such that Lemma 2 (in Appendix D) can be true. From Lemma 2, we prove that SkillWrapper is sound and probabilistically complete.
>
> 3. __Suggestions for ablation study.__ (a) We have studied random exploration baseline as another data collection strategy. However, it’s impossible to remove active data collection entirely, since transition data is necessary for the remaining parts of the system. (b) We believe that the reviewer meant “SkillWrapper-learned operators” by “SkillWrapper-learned skills”, because SkillWrapper does not learn skills. We are unsure about how two sets of operators could be directly combined. We would like to ask if the reviewer could elaborate on this idea or provide a certain reference for us to better understand this question.
>
> ---
>
> __References:__
>
> [1] Konidaris, George, Leslie Pack Kaelbling, and Tomas Lozano-Perez. "From skills to symbols: Learning symbolic representations for abstract high-level planning." Journal of Artificial Intelligence Research 61 (2018): 215-289.
>
> [2] Silver, Tom, et al. "Predicate invention for bilevel planning." Proceedings of the AAAI Conference on Artificial Intelligence. Vol. 37. No. 10. 2023.
>
> [3] Ahmetoglu, Alper, et al. "Deepsym: Deep symbol generation and rule learning for planning from unsupervised robot interaction." Journal of Artificial Intelligence Research 75 (2022): 709-745.
>
> [4] Li, Bowen, et al. "Bilevel Learning for Bilevel Planning." In Proc. R:SS, 2025.
>
> [5] Shah, Naman, Jayesh Nagpal, and Siddharth Srivastava. "From Real World to Logic and Back: Learning Generalizable Relational Concepts For Long Horizon Robot Planning." In Proc CoRL, 2025.

---

> > ### Author Response · Authors · 2025-12-04
> >
> > We have now completed all proposed experiments in the general response, including an ablation study for skill sequence proposal heuristics that could answer __Q3.(a)__ in the original review. The results are incorporated and updated in the manuscript. Please see our latest general response for the details. We believe that we have fully answered and addressed all questions raised by the reviewer.

---

### Official Review · Reviewer_2cbF · 2025-10-31

**Soundness:** 2
**Presentation:** 3
**Contribution:** 3
**Rating:** 6
**Confidence:** 3

**Summary:**

This paper discusses how to learn predicates and operators for given skills. It builds a system like VisualPredicctor that asks LLMs to propose and ground predicates and uses symbolic methods to evaluate and manage the proposed predicates and operators. It also asks LLMs, together with symbolic heuristics, to propose sequences of actions for exploration and data collection. The method is evaluated in several simulated and real-robot environments.

**Strengths:**

This paper shows that generative predicate invention works in real-robot settings.

The paper is generally well-written and easy to understand.

The LLM exploration heuristics are more interesting to me and deserve more space in the main paper, in my understanding. It would be great to study the exploration effectiveness given each combination of the heuristics in practice.

**Weaknesses:**

* The theories are either trivial or missing important strong assumptions in the main context. For example, Theorem 2 relies on i.i.d. samples which is a strong assumption (never true in online-exploration or LLM-exploration settings in practice) and is **not** stated in Theorem 2. With i.i.d. samples, Theorem 2 is true for any method (such as VisualPredicator) that satisifies $\hat Err(\hat M_n) = 0$.
* The predicate-invention method is very similar to VisualPredicator.
* Missing baseline such as VisualPredicator

**Questions:**

* Are there results to compare with VisualPredicator? What's the difference and why?
* Are there results analyzing the effectiveness of various exploration strategies?

---

> ### Author Response · Authors · 2025-11-18
>
> We thank the reviewer for their thoughtful comments on this manuscript and appreciate this chance to clarify the novelty of SkillWrapper. Because the reviewer especially emphasized VisualPredicator [Liang et al., ICLR 2025] as a related prior work, we would like to begin by contrasting it with SkillWrapper. We highlight the following limitations of VisualPredicator:
>
> 1. __Lack of theoretical guarantees:__ A key difference between VisualPredicator and SkillWrapper is that we have proven SkillWrapper to be sound and probabilistically complete, whereas VisualPredicator does not make these theoretical claims.
>
> 2. __Unsoundness of VisualPredicator:__ Beyond the lack of positive proofs of theoretical guarantees for VisualPredicator, we will sketch an argument for the negative case: that VisualPredicator is provably neither sound nor probabilistically complete. A key contribution of VisualPredicator is its “novel objective” that scores a candidate set of predicates (see their Sec. 5.3, pg. 6). This score 1) learns high-level actions (HLAs, i.e., planning operators) for a candidate set of predicates and then 2) computes the classification accuracy of those HLAs. To select a predicate set, VisualPredicator uses greedy best-first search (GBFS) with this score function as a heuristic (Sec. 5.3, pg. 6; Line 13 of Alg. 1, pg. 5). Beginning from the set of provided predicates $\Psi_0$ (more on this in our next bullet), each iteration of GBFS adds one new predicate, eventually returning the highest-scoring predicate set found during search. Although we do not believe* that the score proposed by VisualPredicator would produce sound representations, even if it did, the use of a greedy algorithm precludes any guarantee that the resulting predicate set would maximize the score function.
> In contrast, ***SkillWrapper only invents predicates when necessary to reconcile the symbolic model with observed data, and validates that a proposed predicate improves the model before keeping it. Hence, our method does not require an ad-hoc greedy predicate selection procedure, enabling proofs of theoretical guarantees.***
>
>     - \* We have specific theoretical critiques of the score functions proposed by VisualPredicator (i.e., Eq. 4, App. B.2, pg. 15, and Eq. 5, App. B.3, pg. 16), and we are open to continuing this discussion if the reviewer believes it to be fruitful.
>
> 3. __Engineered pre-defined predicates:__ VisualPredicator relies heavily upon an “initial predicate set $\Psi_0$” given as input to the system (Sec. 5, pg. 5). These predicates define oracle classifiers and include both goal predicates and additional predicates, such as $\mathtt{OnTable}, \mathtt{MachineOn}$ (App. D, pg. 27-28). In contrast, ***SkillWrapper invents predicates from an initially empty predicate set and assumes no handwritten predicates.***
>
> 4. __Pre-specified operator preconditions and effects:__ Further, VisualPredicator is provided with partially specified operators with effects already defined for all five of its evaluation tasks (tasks shown in Fig. 3, pg. 7; provided operators detailed in App. D, pg. 26-29). For one task, the provided operators also specify preconditions (App. D, pg. 29). These strong assumptions are necessary because VisualPredicator explores by planning using its current symbolic model (Sec. 5.1, pg. 5) and ***would not be able to do so if it began with an empty model***. In contrast, ***SkillWrapper neither assumes nor requires human-specified operators because it uses a distinct VLM-based method for active data collection.***
>
> 5. __Human-provided training tasks:__ VisualPredicator is provided with a set of training tasks, each specifying the objects, initial state, and goal. Although the VisualPredicator paper didn’t mention how these goals are represented, its Appendix C.3 (pg. 25) notes: "...we resorted to a simpler goal-count heuristic, which estimates the distance to the goal by counting the number of unsatisfied goals.” We believe it is reasonable to infer that these goals are specified using the provided predicates, making the training tasks equivalent to PDDL-style planning problems. This would explain how the VisualPredicator method can compute abstract plans to “solve the training tasks” (Sec. 5.1, pg. 5). These training tasks, therefore, enable the system designers to guide the system toward abstract states of interest during its exploration. VisualPredicator additionally depends on the pre-defined task goals to trigger predicate invention (condition 2 in Sec. 4, pg. 5). In contrast, ***SkillWrapper performs autonomous active exploration without assuming access to pre-defined PDDL-style problems or oracle classifiers for goal conditions.***

---

> ### Author Response · Authors · 2025-11-18
>
> 6. __Factored, object-centric state with additional input:__ In the formal framework of VisualPredicator, each state includes an RGB image, proprietary information of the robot, and "associated object features, such as 3D object position" (Tasks, Sec. 2, pg. 2). The VLM accesses these features using a Python API (App. A, pg. 14). In doing so, ***VisualPredicator assumes access to a pre-factored, object-centric state with per-object features, whereas SkillWrapper operates exclusively on raw RGB image observations without privileged state information.***
>
> 7. __Lack of real-world experiments:__ Liang et al. (ICLR 2025) only present evaluation results in simulation environments (Experimental Setup, Sec. 6, pg. 7). ***In contrast, we evaluate SkillWrapper and baseline methods on two real-world robot platforms.***
>
> 8. __Lack of theoretical contributions:__ By nature of both inventing predicates to learn symbolic models from observed skill transitions, it is only natural that algorithmic and conceptual similarities arise between SkillWrapper and VisualPredicator. However, we highlight that ***one of our intended contributions is to enable formal comparison between principled methods.*** We hope that the predicate invention research community will appreciate our emphasis on formal guarantees and focus on these properties in future work. More importantly, we hope the community continues to iterate on these ideas, providing a clearer and increasingly principled path toward progress in this area.
>
> The distinctions above make SkillWrapper a more principled and practical system for predicate invention in real-world scenarios. There are also distinctive, albeit inessential, engineering choices made by VisualPredicator that are not employed by SkillWrapper:
>
> 1. VisualPredicator follows prior work in using Set-of-Mark (SoM) prompting [Yang et al., arXiv 2023], where each object in the scene is labeled by overlaying IDs (Evaluating Primitive NSPs, Sec. 3, pg. 4). This helps their system disambiguate object references in environments where natural language is insufficient. We do not apply such an overlay in our robot experiments because it is unnecessary for object disambiguation in our environments and is not a major technical contribution in either paper.
>
> 2. VisualPredicator prompts VLMs for “derived predicates,” building upon existing predicates, "without conditioning on the raw planning data" (Strategy #3, Sec. 5.2, pg. 6). SkillWrapper does not invent derived predicates because it only invents predicates when necessitated by the empirical performance of the current symbolic model on the observed data. We argue that this is a more principled approach and note that our formal framework directly applies to derived predicates, making them a trivial extension.
>
> In sum, although VisualPredicator and SkillWrapper address superficially similar problems, our method relaxes multiple restrictive assumptions made by VisualPredicator, invents predicates and operators from scratch without assuming access to human-engineered abstractions, and can do so thanks to key algorithmic design decisions motivated by our theoretical framework.

---

> ### Author Response · Authors · 2025-11-18
>
> __Weaknesses & Questions:__
>
> 1. - (a) __Assumptions of Theorem 2:__ We mentioned the i.i.d. assumption in full version in Appendix E but not in the main paper due to the page limit, and we thank the reviewer for pointing it out. We agree that the paper is better served by providing the theorem statement with important assumptions in the main body of the paper, and we believe this issue has been resolved in the updated version.
>
>    - (b) **VisualPredicator satisfies $\widehat{Err}(\widehat{\mathcal{M}}_n)=0$.** We do not believe that VisualPredicator satisfies $\widehat{Err}(\widehat{\mathcal{M}}_n)=0$, because its predicate selection score function considers only operator preconditions, but not effects (Eq. 5, App. B.3, pg. 16). Even if VisualPredicator were to satisfy this condition, as mentioned in our detailed comparison, it does so relative to a greedily selected set of predicates which may erroneously discard soundness- or completeness-preserving predicates. This is because GBFS cannot guarantee that the selected predicate set will globally maximize its selection heuristic.
>
>    - (c) **The theories are trivial:** We believe that our theoretical framework is novel and characterizes sufficient conditions for sound and complete predicate invention. This theory informs the design of the SkillWrapper algorithm and provides formal guarantees on the properties of the abstractions learned by the system, helping to explain its strong performance in simulation and on real robots. To the best of our knowledge, existing methods using foundation models for predicate invention (e.g., VisualPredicator) do not claim to guarantee soundness and completeness, and may not provide that guarantee. Our full comparison with VisualPredicator argues why their method does not satisfy the conditions we have defined. However, to help consolidate this discussion, we would appreciate it if the reviewer could clarify specific concerns with our theory and/or proofs, or provide further explanation, e.g., existing works have discussed our exact theory.
>
> 2. __Similar method for predicate invention.__ The two predicate invention methods appear similar at a glance due to the nature of model learning: for any method, learning preconditions will necessarily involve comparing successful and failed transitions to classify skill executability at each state; learning effects will necessarily involve comparing states before and after executing a skill. Unlike VisualPredicator, we would like to highlight that our predicate invention algorithm, including its conditions for invention, prompting strategy, and selection criterion, tightly adheres to our theory. Our full comparison with VisualPredicator clarifies other major distinctions.
>
> 3. __Comparison to VisualPredicator:__ We do not compare against VisualPredicator because SkillWrapper addresses a different, more general class of problem, making a direct fair comparison between the methods impossible. SkillWrapper handles a broader class of problems due to our distinct approaches for active exploration, predicate invention and selection, and operator learning. See our full comparison for further details on the distinctions (e.g., we do not assume an initial predicate set, initial operator preconditions and effects, or a factored object-centric input state) between the two approaches.
>
> 4. __Effectiveness of various exploration strategies.__ We have included random exploration as a baseline to demonstrate the value of active data collection, and we are actively running ablation studies that remove the exploration heuristics. We will update with the results as soon as possible.
>
> 5. __Soundness.__ We would like to ask if the reviewer has other concerns that affect the overall judgement on the soundness of the paper beyond those addressed above.
>
> ---
>
> __Citations:__
>
> Y. Liang et al., “VisualPredicator: Learning Abstract World Models with Neuro-Symbolic Predicates for Robot Planning,” presented at the Thirteenth International Conference on Learning Representations, Jan. 2025. Available: https://openreview.net/forum?id=QOfswj7hij.
>
> J. Yang, H. Zhang, F. Li, X. Zou, C. Li, and J. Gao, “Set-of-Mark Prompting Unleashes Extraordinary Visual Grounding in GPT-4V,” Nov. 06, 2023, arXiv: arXiv:2310.11441. doi: 10.48550/arXiv.2310.11441.

---

> > ### Author Response · Authors · 2025-12-04
> >
> > We have now completed all proposed experiments in the general response, including an ablation study for skill sequence proposal heuristics that could answer __Q2__ in the original review. The results are incorporated and updated in the manuscript. Please see our latest general response for the details. We believe that we have fully answered and addressed all questions raised by the reviewer.

---

### Official Review · Reviewer_3RUj · 2025-10-31

**Soundness:** 3
**Presentation:** 3
**Contribution:** 3
**Rating:** 6
**Confidence:** 3

**Summary:**

The paper introduces SKILLWRAPPER, a framework for generative predicate invention that learns human-interpretable symbolic models of black-box skills for long-horizon planning from RGB images only. The method formalizes when and how to invent new predicates so that the resulting abstraction yields provably sound and (probabilistically) complete PDDL operators for planning. Practically, the system (i) actively gathers data by proposing skill sequences, (ii) invents predicates when failures/successes are indistinguishable under the current vocabulary, and (iii) learns operators via associative model learning with type hierarchies and periodic predicate re-evaluation (Algs. 1–2). Experiments in the Robotouille grid-world kitchen and on two real robot setups (Franka Panda; bimanual Kuka) show higher solve rates and lower planning budgets than VLM prompting and random exploration, and competitive performance against expert-authored operators; results include generalization to richer environments and handling “impossible” tasks. The novelty lies in a formal theory of predicate invention tailored to skill abstraction with guarantees, plus a concrete system that uses a foundation model both to propose predicates and to classify their truth values directly from images.

**Strengths:**

1. Originality: Provides a formal theory for generative predicate invention specifically for skill abstraction, with explicit conditions tied to precondition/effect indistinguishability (Sec. 3.2), addressing a gap in prior ad-hoc predicate generation.
2. Real-world evaluation: Includes two real-robot settings (Franka; bimanual Kuka), with generalization across object/skill subsets and learning curves that surpass baselines as predicates accumulate (Figs. 3–5, Table 2).
3. Proves soundness of learned operators and probabilistic completeness relative to a finite hypothesis class (Theorems 1–2), linking learning criteria directly to planning guarantees (Sec. 3.4).

**Weaknesses:**

1. The approach assumes accurate truth-value predictions from a foundation model (Sec. 4.1), which might not always work in real world.
2. Results are averaged over three runs, with no error bars or significance tests (Sec. 4).
3. Real-robot sections assume deterministic skills and fully observable states (Sec. 4), which may not hold in cluttered, partially observable settings.
4. Lack of compute budgets (GPU/CPU hours), inference costs for predicate evaluations (the cost of GPT-5).

**Questions:**

1. What is the per-predicate truth-value accuracy of the VLM classifier, and how does plan success degrade under controlled label noise?
2. Can you report mean and std (or confidence intervals) over >= 5 seeds for Table 1 and Table 2?
3. What are the compute budgets (GPU/CPU hours)? How much do you spend on GPT-5 for evaluation and predicate generation?
4. Could you list some scenarios of failure as case studies?
5. In the real-robot setups, how often did image viewpoint changes alter predicate truth judgements? Any mitigations (multi-view, temporal smoothing)?

---

> ### Author Response · Authors · 2025-11-18
>
> We would like to thank the reviewer for the insightful comments. We appreciate the constructive comments that would strengthen the paper.
>
> __Weaknesses:__
>
> 1. __Assumes accurate truth-value predictions.__ We agree that the accuracy of VLMs is an important factor. However, we would like to clarify that our method does not require the VLM to be strictly accurate.  Please see General Response 3 for more details. In the two sets of experiments on real robots, we have demonstrated that SkillWrapper can reliably invent predicates and learn operators from raw images to solve abstract planning problems in the real world.
>
> 2. __Result statistics.__ We have updated the manuscript with the statistics, and we will update it again once all the suggested experiments are complete.
>
> 3. __Assuming deterministic skill and full observability.__ Please see General Response 1.
>
> 4. __Compute budget and inference costs.__ We addressed this problem by answering Q3, and we have updated these experimental details in the paper (Appendix H).
>
> __Questions:__
>
> 1.  - (a) __Per-predicate truth value accuracy.__ We have provided the per-predicate accuracy breakdown in Appendix H.1, and we will update more results in the appendix when all experiments in General Response 2 are done.
>     - (b) __How plan success degrades under noise.__ We agree that understanding performance degradation under noise is valuable. However, we emphasize that our real-world experiments implicitly evaluate this robustness. In transferring from Bimanual Kuka to Franka, the VLM classification accuracy dropped significantly (from 98.5% to 86.7%), representing a realistic, non-trivial noise injection. Despite this degradation, our system maintained satisfying planning performance. We believe this real-world stress test is more indicative of practical robustness than synthetically degrading the VLM further. Given this evidence, we prioritized the additional experiments listed in the General Response (2.1).
>
> 2. __Mean and std for 5 seeds.__ We have updated the result tables and Figure 6 with standard deviations, and we are actively running the remaining seeds. We will update more results as soon as they become available. Given the time limit and the amount of additional experiments that we are asked for, we will only be able to run the Robotouille experiments (Table 1). We hope this is understandable to the reviewer.
>
> 3. __Computing budget and GPT-5 API call.__ We did not use GPUs. We have measured the average planning time, and it is mostly negligible (~0.055 seconds per planning problem). We updated more details in Appendix H.1. For running the experiments, we made roughly 9300 calls to GPT-5, which cost $96.59 in total. We have updated this detail in Appendix H.2.
>
> 4. __Failure cases.__  We have provided case studies in Appendix F.2 (Previously E.2), and we have updated more details that may also interest the reviewer.
>
> 5. __Viewpoint changes affect accuracy.__ In the Franka experiments, the learning process and evaluation use visual observations from a fixed camera in front of the robot. In the bimanual Kuka experiments, we use visual observations from the robot’s egocentric view. We have analyzed the reliability of VLM for both experiments in Appendix G. From the results, we observe that the misclassifications are usually caused by certain objects, background, or lighting conditions, while it is generally robust to the viewpoint changes as long as the state is fully observable. To better understand VLMs’ reliability, we are running more experiments (please see General Response 2), and we will report the results as we complete them. We did not implement any method on top of prompting the foundation model with raw images, because we don’t consider improving the reliability of VLMs as one of our contributions, while we do agree that incorporating the suggested approaches can likely make the system more robust in practice.

---

> > ### Comment · Reviewer_3RUj · 2025-11-25
> >
> > Thank you for the detailed response. I have one more question. I am curious about the use of GPT-5. Why do authors choose GPT-5? Is there any other choice about the models? For example, what does the result change using cheap opensource models such as Qwen3-32B, GLM-4.6, kimi-k2? I understand that the rebuttal time is going to end, so it is not compulsory to run all models.

---

> ### Author Response · Authors · 2025-11-25
>
> We thank the reviewer for their further interest. We believe the capabilities of the foundation model underlying SkillWrapper, including skill sequence generation, predicate invention, and predicate classification, are critical to our method’s overall performance. Therefore, we have consistently used the best available vision-language model (VLM) throughout this project. To our knowledge, this has been a variant of GPT-4o or GPT-5, specifically the OpenAI checkpoints “gpt-4o-2024-08-06,” “gpt-4o-2024-11-20,” “o1” (text-only), and “gpt-5.”
>
> We observed qualitative performance improvements as newer models were released. For instance, older checkpoints would invent the predicate $\mathtt{ItemIsPickupable(item)}$ for the skill $\mathtt{PickUp(item)}$ even when we explicitly instructed the model not to do so, whereas we did not observe this problem after switching to newer checkpoints. This finding reinforces our belief that SkillWrapper can benefit from continued advances in the commonsense reasoning capabilities of large pretrained models.
>
> We agree that exploring open-source or open-weight models, even if less capable, could provide insights into how SkillWrapper’s performance depends on VLM capabilities and would improve the reproducibility of our approach. However, we note that the specific models mentioned by the reviewer (Qwen3-32B, GLM-4.6, kimi-k2) are large language models, not VLMs. That said, open-source VLM variants exist for some of these model families: Qwen3-VL-32B-Instruct (released Oct. 2025), GLM-4.5V (released Aug. 2025), and Kimi-VL (released Apr. 2025). In this work, we did not evaluate additional models beyond those mentioned above, but we view this as a promising direction for future work.

---

> > ### Comment · Reviewer_3RUj · 2025-11-26
> >
> > Thank you for the reminder from the authors. I meant to express that using some open-source VLMs (such as those mentioned by the authors) to conduct some ablations would be better.

---

> > > ### Author Response · Authors · 2025-11-27
> > >
> > > We thank the reviewer for their clarification. We first note that ablation studies across foundation models are not standard practice in the literature on symbolic model learning, where the focus is typically on the abstraction methodology rather than variability in the VLM backbone [1, 2]. Nonetheless, we agree that understanding SkillWrapper’s dependence on the capabilities of the underlying VLM is important.
> > >
> > > Given the constraints of the rebuttal period, we propose two targeted comparisons between **Qwen3-VL-235B** and **GPT-5**:
> > > - Predicate classification: Measure classification accuracy for a subset of the predicates previously invented during our real-world robot experiments, using a subset of images from those experiments.
> > > - Predicate invention: Qualitatively compare invented predicates using a curated set of contrastive image pairs from the same experiments.
> > >
> > > These experiments directly evaluate the core VLM reasoning capabilities SkillWrapper relies upon, although they do not test how these capabilities affect the overall performance of our method. Answering that question would require running one or more additional full-system experiments, which we do not believe is feasible given the window for rebuttal responses.
> > >
> > > We anticipate that our current experiments will conclude within the next 24 hours, after which we will run the VLM comparison and post the results as soon as they are available. We appreciate the reviewer’s attention to the rebuttal timeline.
> > >
> > > [1] Y. Liang et al., “VisualPredicator: Learning Abstract World Models with Neuro-Symbolic Predicates for Robot Planning,” ICLR. 2025. Available: https://openreview.net/forum?id=QOfswj7hij.
> > >
> > > [2] Liu et al., “Learning Planning Abstractions from Language,” ICLR. 2024. Available: https://openreview.net/forum?id=3UWuFoksGb

---

> > > > ### Author Response · Authors · 2025-11-30
> > > >
> > > > We have now completed all suggested experiments in the general response and the open-source VLM study that was later added by the reviewer. The results are incorporated and updated in the manuscript. Please see our latest general response for the details. We believe that we have fully answered and addressed all questions raised by the reviewer.

---

### Author Response · Authors · 2025-11-18
**General Response**

We thank all the reviewers for their thoughtful and actionable comments. We are glad that reviewers found our work original, “generally well-written”, “easy to understand”, clear, and significant. We are happy that our paper was able to convey to reviewers our core contribution on developing a theory that learns symbolic models for skills while providing guarantees of completeness and soundness of learned models.

We would now like to respond to some of the questions raised by the reviewers. Here, we post a common response that answers questions raised by more than one reviewer and then respond to each reviewer separately to address reviewer-specific questions.

__1. Deterministic skills and fully observable environment. (Reviewer 3RUj, eiUT)__

We make two key assumptions: deterministic skills and a fully observable environment. These assumptions are commonly made in prior and concurrent work in predicate invention for model learning [1, 2, 3, 4, 5, 6, 7]. We have updated the paper to clarify this. While we did not relax these assumptions in the paper, we would like to highlight our contribution on the formal theory for provably sound and complete skill abstraction, the SkillWrapper algorithm for principled generative predicate invention, and implementation and evaluation on real robots. To our knowledge, SkillWrapper is the first method to demonstrate generative predicate invention for skill abstraction on real robots using RGB image observations.

We believe that relaxing these assumptions without losing formal guarantees is a promising direction for future work, potentially by using approaches such as SLAM that can incorporate spatial memory. However, these extensions are out of scope for this paper, and we would like to defer stochastic and partially observable settings to future work.

__2. Additional Experiments.__

We thank reviewers for their constructive feedback to improve the quality of the papers. We have compiled a list of experiments that one or more reviewers have suggested. We will try our best to finish the suggested experiments. However, due to time limitations, we may not be able to complete all the experiments promptly. Therefore, here is an ordered list of experiments (higher priority = higher in the order) that we would try to conduct.

1. VLM classification test on viewpoint/lighting (Franka, Kuka)
2. Table 1 with 5 individual runs (Robotouille)
3. Ablation for skill sequence proposal heuristics (Robotouille)
4. VLM classification test on minor domain shift (Franka, Kuka)

We note that additional simulation results will likely be completed first, while experiments on the robots will take more time. We would try to report the results as we have them. Please let us know if you have different opinions on the proposed ordering of these additional experiments.

__3. Classification noise from VLMs. (Reviewer 3RUj, eiUT)__

We agree that VLM classification accuracy is an important factor in our system. However, this highlights the importance of having a correct and complete theory, as our proposed method uses theoretically motivated conditions to identify reliable predicates. We emphasize that our contributions do not center on the current visual capabilities of VLMs, but rather on how principled algorithms can use these potentially noisy classifiers to produce sound representations for planning. Additionally, as VLMs continue to evolve, SkillWrapper offers a modular framework that can directly benefit from their advancement.

In practice, we incorporate a per-predicate scoring function to account for this classification noise (see Alg. 6). Specifically, this scoring function is used to verify that any proposed predicate empirically improves the accuracy of the symbolic model over the available data. If the VLM cannot reliably classify the condition for a predicate, that predicate is unlikely to improve the symbolic model and would therefore not be kept. This process may discard “near-reliable” predicates, but in doing so, it maintains the soundness and probabilistic completeness properties of our approach.

We also conducted analyses on VLMs’ accuracy in both robotic experiments, since multiple reviewers expressed their concerns about the real-world reliability of the VLMs. From the result, we can conclude that the VLMs are reliable enough for the predicate classification task. Despite specific predicates failing to get reliably correct results on certain objects or due to lighting conditions, the overall accuracy achieves 86.7% in Franka and 98.5% on bimanual Kuka (more details could be found in Appendix G). We believe a more advanced vision pipeline or pre-trained model could potentially further improve the performance of SkillWrapper, which we will leave for future work.

---

> ### Author Response · Authors · 2025-11-18
>
> __4. Changes in the updated manuscript__
>
> We have updated the manuscript to incorporate reviewers feedback and improve clarity. The changes in the main paper are:
> - Standard deviation for all result tables and figure 6.
> - Theorem 2 with necessary assumptions.
> - More detailed description of the system predicate baseline.
> - Predicate invention example (figure 2).
> - Consistent notation throughout the paper.
>
> The changes in the appendix are:
> - Derivation of upper bound of |H| and sample complexity (Appendix E).
> - VLM reliability study (Appendix G).
> - More detailed case study (Appendix F.2).
> - Example tasks for each experiment (Appendix F.3).
> - Computation resource, API cost, and hyperparameters (Appendix H.1, H.2, H.3).
> - Clearer pseudo code.
>
> ---
>
> Meanwhile, as we are actively running the suggested experiments, we would like to ask if the reviewers have follow-up questions, and we would like to continue these valuable conversations with a scientific mind.
>
> Lastly, if our responses have addressed your concerns expressed in questions and weaknesses, we would greatly appreciate it if you would consider adjusting your scores. Thank you!
>
> ---
>
> __References__
>
> [1] J. Huang, A. Tao, R. Marco, M. Bogdanovic, J. Kelly, and F. Shkurti, “Automated Planning Domain Inference for Task and Motion Planning,” in 2025 IEEE International Conference on Robotics and Automation (ICRA), IEEE, 2025, pp. 12534–12540. Accessed: Sept. 13, 2025. [Online]. Available: https://ieeexplore.ieee.org/abstract/document/11127817/
>
> [2] M. Han, Y. Zhu, S.-C. Zhu, Y. N. Wu, and Y. Zhu, “InterPreT: Interactive Predicate Learning from Language Feedback for Generalizable Task Planning,” in Robotics: Science and Systems (RSS), 2024
>
> [3] E. Umili, E. Antonioni, F. Riccio, R. Capobianco, D. Nardi, and G. D. Giacomo, “Learning a Symbolic Planning Domain through the Interaction with Continuous Environments”. in Bridging the Gap Between AI Planning and Reinforcement Learning (PRL) Workshop at ICAPS 2021.
>
> [4] T. Silver, R. Chitnis, J. Tenenbaum, L. P. Kaelbling, and T. Lozano-Pérez, “Learning Symbolic Operators for Task and Motion Planning,” in 2021 IEEE/RSJ International Conference on Intelligent Robots and Systems (IROS), Sept. 2021, pp. 3182–3189. doi: 10.1109/IROS51168.2021.9635941.
>
> [5] T. Silver, R. Chitnis, N. Kumar, W. McClinton, T. Lozano-Pérez, L. Kaelbling, and J. Tenenbaum (2023). Predicate Invention for Bilevel Planning. In Proceedings of the AAAI Conference on Artificial Intelligence, 37(10), 12120-12129. https://doi.org/10.1609/aaai.v37i10.26429
>
> [6] A. Athalye, N. Kumar, T. Silver, Y. Liang, T. Lozano-Pérez, and L. P. Kaelbling, “Predicate Invention from Pixels via Pretrained Vision-Language Models,” Dec. 31, 2024, arXiv: arXiv:2501.00296. doi: 10.48550/arXiv.2501.00296.
>
> [7] Y. Liang et al., “VisualPredicator: Learning Abstract World Models with Neuro-Symbolic Predicates for Robot Planning,” in Thirteenth International Conference on Learning Representations (ICLR), Oct. 2024. Accessed: Nov. 14, 2025. [Online]. Available: https://openreview.net/forum?id=QOfswj7hij

---

> > ### Author Response · Authors · 2025-11-30
> >
> > We thank all reviewers again for their constructive comments. We have finished all experiments listed above, and we have also incorporated the open-source VLM study later suggested by reviewer 3RUj. The results are incorporated and updated to the manuscript.
> >
> > The key changes to the revised manuscript are summarized below:
> >
> > __Changes in the main paper__:
> > - All results in Table 1 are now averaged over five individual runs.
> > - The “No Heuristic” baseline has been incorporated into Table 1.
> >
> > __Changes in the appendix__:
> > - Analysis of the “No Heuristic” baseline has been added to Appendix F.2.
> > - Real-world VLM robustness studies has been added to Appendix G.2, covering viewpoint changes, lighting condition changes, and domain shift.
> > - An open-source VLM study (comparing GPT-5 with Qwen3-VL-235B) has been added to Appendix G.3.

---

### Author Response · Authors · 2025-12-04
**Summary**

Our paper makes three core contributions: (1) a formal theory for provably sound and complete skill abstraction, (2) the SkillWrapper algorithm for principled generative predicate invention, and (3) implementation and evaluation on real robots. We are glad that the reviewers recognized the strengths of our paper in their reviews, highlighting the value of these contributions by noting our **real robot experiments** as crucial (reviewer **3RUj**, **2cbF**, **FNJ2**, **eiUT**), the **formal theory** as a significant advance over prior ad-hoc methods for predicate generation (reviewer **3RUj**), and the paper overall as a well-written and **important step toward integrating FM-driven perception with symbolic planning in robotics** (reviewer **eiUT**).

To assist the AC in understanding how we have addressed the reviewers’ comments, we would like to summarize the outcomes from our responses during the rebuttal period. We appreciate the reviewers’ original comments and hope that this summary will assist the AC in synthesizing their meta-review.

---

__General Response:__

  1. __Assumptions.__ Reviewers **2cbF** and **eiUT** raise concerns regarding our assumptions of deterministic skills and fully observable environments. In our general response, we clarified that this is the typical problem setting in prior and concurrent work on predicate invention and symbolic model learning. Although we did not relax these assumptions, we have contributed a novel theoretical framework for learning sound and probabilistically complete abstractions in these settings. To our knowledge, SkillWrapper is also the first work to demonstrate and evaluate predicate invention on real robots using only RGB image observations.
   2. __Additional Experiments.__ In our original general response, we summarized and prioritized a list of additional experiments proposed by the reviewers. In the summaries below, we clarify the additional experiments we have conducted during the rebuttal period as relevant to each reviewer’s comments.

---

__Reviewer 3RUj:__

Based on the last message from the reviewer, we believe that we have adequately addressed the concerns raised in their initial review. In their last message, they asked one more question related to using open-source VLMs as alternatives to GPT-5. To address this question, we have conducted preliminary experiments and updated the manuscript with the results in Appendix G.3.

- __W2, W4, Q1, Q3, Q4:__ We have added all additional details, such as error bars, compute budgets, per-predicate VLM classification accuracy, etc., requested by the reviewer to the manuscript.

- __W1, W3:__ As clarified in our original response to the reviewer’s Q1, our method does not require that the foundation model provide perfect classification accuracy. The reviewer’s other concerns are addressed in the above “Assumptions” section of this response.

- __Q1, Q5:__ We have conducted additional experiments, including five individual runs of the full system and a comprehensive study of the predicate classification accuracy of open-source VLMs using real-world robot data. We did not conduct the controlled label noise experiments proposed by the reviewer, as our current robot experimental results already demonstrate the performance of SkillWrapper under degraded classification accuracy. We believe that the reviewer was satisfied with this based on their reply.

---

__Reviewer 2cbF:__

The reviewer is primarily concerned with the distinction between SkillWrapper and VisualPredicator, a related prior approach to VLM-based predicate invention for model learning. We therefore highlight the differences between our approach and VisualPredicator, which include our theoretical contributions and formal guarantees, our relaxation of multiple restrictive assumptions made by VisualPredicator (e.g., engineered predefined predicates; partially human-specified operators; human-provided training tasks for data collection; and a factored, object-centric state with additional inputs beyond raw images), and our real-world experimental results. Our full response to the reviewer further elucidates these distinctions.

- __W1:__ We have presented evidence that VisualPredicator does not satisfy the condition in our Theorem 2, and we clarified the assumptions of the theorem by moving its full statement from Appendix E to the main body of the paper. We therefore have addressed all the reasons provided by the reviewer when describing our theory as “trivial.”

- __W2, W3, Q1:__ We addressed these concerns in our original response to the reviewer, and briefly summarized the differences between SkillWrapper and VisualPredicator at the beginning of this reviewer-specific rebuttal summary.

- __Q2:__ We have a random exploration baseline originally, and we later added another baseline suggested by Reviewer __eiUT__. We believe that these experiments demonstrate the effectiveness of the proposed exploration strategy.

---

> ### Author Response · Authors · 2025-12-04
>
> __Reviewer FNJ2__
>
> We believe that our original response to the reviewer addressed their concerns by 1) clarifying our central contribution and several important details of our paper that we believe the reviewer overlooked, and 2) pointing to specific existing content in the paper to answer their questions. We were unable to interpret some of their comments without further information or clarification.
>
> - __W1:__ We clarified that the expert operator baseline is a conceptual upper bound. A major motivation for SkillWrapper, as with all work in predicate invention for symbolic abstraction learning, is to facilitate model learning without requiring human experts to manually engineer planning operators.
>
> - __W2, Q2:__ We pointed the reviewer to the corresponding sections of the paper and briefly summarized the relevant details.
>
> - __W3:__ We clarified to the reviewer that the two suggested benchmarks are actually for skill learning with reinforcement learning (RL), and IsaacGym is already deprecated. Neither of these is appropriate for our problem setting, in which an agent is provided with pre-defined skills and must learn symbolic abstractions to facilitate long-horizon problem-solving with those skills.
>
> - __W4:__ We could not respond to this question effectively because the reviewer did not provide examples of alternative baselines or clarify how our presentation fails to convey the core novelty of our work. Nonetheless, we pointed the reviewer to the paragraphs that summarize our core contributions and novelty.
>
> - __Q1:__ We explained that ViLa, a VLM-based method for multimodal reasoning, does not use the Planning Budget metric because ViLa does not perform symbolic planning
>
> - __Q3:__ We tried to interpret the reviewer’s question. We hypothesized that the reviewer meant alternative methods of data collection in the first half of the question, because removing data collection completely would result in our method having no data to learn from. Under this interpretation, we had a random exploration baseline originally, and during the rebuttal, we added another baseline ablation: SkillWrapper, with the two exploration heuristics removed. We believe these answer the first half of the question. However, the second half of the question did not make sense to us, and we requested additional clarification from the reviewer.

---

> ### Author Response · Authors · 2025-12-04
>
> ---
>
> __Reviewer eiUT__
>
> The reviewer categorized their questions, so we indexed them in our original response for clarity. We use the same indexing scheme here for consistency.
>
> - __W1.(a):__ The reviewer is concerned about the completeness of SkillWrapper being only empirical. We clarified that we defined a complete model in a theoretical way, while the completeness of SkillWrapper has to be validated empirically due to its data-driven nature. Moreover, the completeness of SkillWrapper already guarantees that the learned model is sound.
>
> - __W1.(b), W1.(d), W3.(e), Q2:__ We responded to the concern regarding the assumption of full observability in our general response. We agree that our theory does not incorporate noise, though we would like to note that we are nonetheless the first method with formal guarantees of soundness and probabilistic completeness for this problem setting.
>
> - __W1.(c), Q1:__ Per the suggestion of the reviewer, we have derived an upper bound on $|H|$ and added the derivation to Appendix E.
>
> - __W1.(e):__ We have clarified that our system can learn predicates that involve objects not present in the arguments of a skill instance. For example, suppose a robot approaches a table `t1` by executing a skill instance `GoTo(t1)`. If a box `b1` is resting on the table, perhaps represented by the predicate `On(b1, t1)`, and this fact is potentially relevant for a later skill (e.g., `Pick(?box)`), SkillWrapper may learn an operator representing the predicate `Reachable(?box)` is a conditional effect of `GoTo(?table)`. In this case, the relevant operator would have `On(?box, ?table)` as a precondition, expressing that “When the robot goes to a table, it can reach a box on the table.” Crucially, the grounded operator can potentially include objects not present in the corresponding skill instance.
>
> - __W2.(a):__ We clarified that re-evaluation and collecting additional data are not alternatives to each other. We agree that knowing when to stop collecting data is important, but we believe that this is a difficult problem that arises in any active learning context. Solving it here would require additional assumptions beyond those made in this work. We therefore view this question as an interesting direction for future work.
>
> - __W3.(a):__ We argued that “System Predicates” is a valuable baseline and that its poor performance on more complex problems is not due to an inherent lack of expressivity. We have updated the paper with a more detailed description to clarify this baseline.
>
> - __Q4:__ We addressed the assumption of stochastic skills in the general response and have clarified that our system handles “side-effectful skills” by design. We are unsure exactly what the reviewer meant by “context predicates” and “effect predicates,” but our system does not treat predicates used as conditional effects any differently from predicates used as typical effects or preconditions. Conditional effects, to the extent that they are differentiated, are identified algorithmically during operator learning.
>
> - __Q5:__ We do not study the cross-platform transferability of abstractions learned by SkillWrapper. There is potential, as the learned predicates classify human-interpretable conditions given RGB images, but we consider this to be out of scope in this work.
>
> - __Q7:__ We have added most metrics requested by the reviewer to the paper. We did not measure plan optimality gaps, as our paper does not study the optimality of the learned models, and our problem settings do not have optimal plans readily available.

---

> ### Author Response · Authors · 2025-12-04
>
> Because the reviewer proposed numerous additional experiments, we list those here using the same indexing:
>
> - __W2.(c), Q3.(c):__ We believe that extensively tuning standard hyperparameters is interesting but not central to our contributions. In practice, trying one of the three initial values for $h$ usually yields good results. Additionally, $h$ mostly affects the number of interactions required for operator learning, but not the final learned operators.
>
> - __W2.(d), Q3.(a):__ Our original response pointed the reviewer to our random exploration baseline. During the rebuttal period, we conducted the proposed ablation experiment by removing the exploration heuristics from SkillWrapper. These results have been added to Table 1.
>
> - __W3.(b):__ We did not find the prior work described as “neurosymbolic predicate learners” by the reviewer, and we have highlighted in our responses to Reviewer __2cbF__ why related prior work, such as VisualPredicator, is incomparable with SkillWrapper. We could not further address this question because no references were provided.
>
> - __W3.(c):__ We updated the result table (Table 1) with 5 runs, and we have added standard deviations to all results in all tables in the experiments section.
>
> - __W3.(d), Q6:__ As suggested by the reviewer, we conducted VLM robustness studies using a held-out set of real-robot data with labelled ground truth predicate classifications. These data included variations in viewpoint, lighting, and domain shifts.
>
> - __Q3.(b):__ We view predicate re-evaluation, which re-scores predicates as new data is collected, as a necessary component of our system that is a standard practice in the literature. We do not think that ablation studies omitting this component would be fruitful.

---

### Meta-Review · Area_Chair_Q927 · 2025-12-23

**Summary:**

The submission introduces a method that leverages foundation models to collect robot data and learn human-interpretable, plannable representations of skills, leveraging a new idea named generative predicate invention.  Reviewers liked the idea but shared concerns about the strong assumptions, limited evaluation in both the simulation and the real world, and missing baselines.

**Reviewer Concerns:**

While some presentation and problem-formulation issues may have been addressed, the AC found that most concerns, including the strong assumption, the limited comparisons, and the insufficient evaluation of more complex tasks, remained unaddressed.  While environments like IsaacGym or MetaWorld may not be suitable, benchmarks such as BEHAVIOR can be used to evaluate the proposed method in much more practical setups.

**Reviewer Scores:**

The reviewers might converge to borderline scores, such as 6, 6, 4, 6.

---

### Decision · Program_Chairs · 2026-01-26

Reject